# Causal Bandits with Unknown Graph Structure

**Yangyi Lu**
Department of Statistics
University of Michigan
yylu@umich.edu

**Amirhossein Meisami**
Adobe Inc.
meisami@adobe.com

**Ambuj Tewari**
Department of Statistics
University of Michigan
tewaria@umich.edu

## Abstract

In causal bandit problems, the action set consists of interventions on variables of a causal graph. Several researchers have recently studied such bandit problems and pointed out their practical applications. However, all existing works rely on a restrictive and impractical assumption that the learner is given full knowledge of the causal graph structure upfront. In this paper, we develop novel causal bandit algorithms without knowing the causal graph. Our algorithms work well for causal trees, causal forests and a general class of causal graphs. The regret guarantees of our algorithms greatly improve upon those of standard multi-armed bandit (MAB) algorithms under mild conditions. Lastly, we prove our mild conditions are necessary: without them one cannot do better than standard MAB algorithms.

## 1  Introduction

A multi-armed bandit (MAB) problem is one of the classic models of sequential decision making (Auer et al., 2002; Agrawal and Goyal, 2012, 2013a). Statistical measures such as regret and sample complexity measure how fast learning algorithms achieve near optimal performance in bandit problems. However, both regret and sample complexity for MAB problems necessarily scale with the number of actions without further assumptions. To address problems with a large action set, researchers have studied various types of structured bandit problems where additional assumptions are made on the structure of the reward distributions of the various actions. Algorithms for structured bandit problems exploit the dependency among arms to reduce the regret or sample complexity. Examples of structured bandit problems include linear bandits (Abbasi-Yadkori et al., 2011; Agrawal and Goyal, 2013b), sparse linear bandits (Abbasi-Yadkori et al., 2012), and combinatorial bandits (Cesa-Bianchi and Lugosi, 2012; Combes et al., 2015).

In this paper, we study a different kind of structured bandit problems: *causal bandits*. In this setting, actions are composed of interventions on variables of a causal graph. Many real world problems can be modeled via causal bandits. In healthcare applications, the physician adaptively adjusts the dosage of multiple drugs to achieve some desirable clinical outcome (Liu et al., 2020). In email campaign problems, marketers adjust for features of commercial emails to attract more customers and convert them into loyal buyers (Lu et al., 2019; Nair et al., 2021). Genetic engineering also involves direct manipulation of one or more genes using biotechnology, such as changing the genetic makeup of cells to produce improved organisms (Wikipedia contributors, 2021). Recently there has been a flurry of works (Lattimore et al., 2016; Sen et al., 2017; Lee and Bareinboim, 2018; Lu et al., 2019; Lee and Bareinboim, 2019; Nair et al., 2021) on causal bandits that show how to achieve simple regret or cumulative regret not scaling with the action set size.

However, a major drawback of existing work is that they require significant prior knowledge. All existing works require that the underlying causal graph is given upfront. Some regret analysis works even assume knowing certain probabilities for the causal model. In practice, they are all strong assumptions.

In this paper, our goal is to develop causal bandit algorithms that 1) do not require prior knowledge of the causal graph and 2) achieve stronger worst-case regret guarantees than non-causal algorithms such as Upper Confidence Bound (UCB) and Thompson Sampling (TS) whose regret often scales *at least polynomially* with the number of nodes $n$ in the causal graph. Unfortunately, this goal cannot be achieved for general causal graphs. Consider the causal graph consists of isolated variables and the reward directly depends on one of them. Then in the worst case there is no chance to do better than standard algorithms since no meaningful causal relations among variables can be exploited. In this paper, we study what classes of causal graphs on which we can achieve the goal.

**Our Contributions.** We summarize our contributions below.

1. We first study causal bandit problems where the unknown causal graph is a directed tree, or a causal forest. This setting has wide applications in biology and epidemiology (Greenewald et al., 2019; Burgos et al., 2008; Kontou et al., 2016; Pavlopoulos et al., 2018). We design a novel algorithm **C**entral **N**ode UCB (CN-UCB) that *simultaneously exploits* the reward signal and the tree structure to efficiently find the direct cause of the reward and then applies the UCB algorithm on a reduced intervention set corresponding to the direct cause.

2. Theoretically, we show under certain identifiability assumptions, the regret of our algorithm only scales *logarithmically* with the number of nodes $n$ in the causal graph. To our knowledge, this is the first regret guarantee for unknown causal graph that provably outperforms standard MAB algorithms. We complement our positive result with lower bounds showing the indentifiability assumptions are necessary.

3. Furthermore, we generalize CN-UCB to a more general class of graphs that includes causal trees, causal forests, proper interval graphs, etc. Our algorithm first constructs undirected clique (junction) trees and again simultaneously exploits the reward signal and the junction-tree structure to efficiently find the direct cause of the reward. We also extend our regret guarantees to this class of graphs.

In many scenarios, our algorithms do *not* recover the full underlying causal graph structure. Therefore, our results deliver the important conceptual message that *exact causal graph recovery is not necessary in causal bandits* since the main target is to maximize the reward.

## 2 Related work

The causal bandit framework was proposed by Lattimore et al. (2016) and has been studied in various settings since then. In the absence of confounders, Lattimore et al. (2016) and Sen et al. (2017) studied the best arm identification problem assuming that the exact causal graph and the way interventions influence the direct causes of the reward variable are given. Lu et al. (2019) and Nair et al. (2021) proposed efficient algorithms that minimize the cumulative regret under the same assumptions. When confounders exist, Lee and Bareinboim (2018, 2019) developed effective ways to reduce the intervention set using the causal graph before applying any standard bandit algorithm, such as UCB. Even though the performance of above works improved upon that of standard bandit MAB algorithms, they all make the strong assumption of knowing the causal graph structure in advance.

In our setting, the underlying causal graph is not known. Then a natural approach is to first learn the causal graph through interventions. There are many intervention design methods developed for causal graph learning under different assumptions (He and Geng, 2008; Hyttinen et al., 2013; Shanmugam et al., 2015; Kocaoglu et al., 2017; Lindgren et al., 2018; Greenewald et al., 2019; Squires et al., 2020). However, this approach is not sample efficient because it is not necessary to recover the full causal graph in order to maximize rewards. de Kroon et al. (2020) tackled this problem with unknown graph structure based on separating set ideas. However, this implicitly requires the existence of a set of non-intervenable variables that d-separate the interventions and the reward. Moreover, their regret bound does not improve upon non-causal algorithms such as UCB. In this paper, we take a different approach which uses the reward signal to efficiently learn the direct causes of the reward.

Our approach is inspired by Greenewald et al. (2019) which proposed a set of central node algorithms that can recover the causal tree structure within $O(\log n)$ single-node interventions. Squires et al. (2020) also extended the central node idea to learn a general class of causal graphs that involves constructing junction trees and clique graphs. Following these works, our causal bandit algorithms

also adaptively perform interventions on central nodes to learn the direct causes of the reward variable. However, our algorithms differ from theirs because we also take the reward signal into account.

## 3 Preliminaries

In this section, we follow the notation and terminology of Lattimore et al. (2016); Greenewald et al. (2019) for describing causal models and causal bandit problems.

### 3.1 Causal Models

A causal model consists of a directed acyclic graph (DAG) $D$ over a set of random variables $\mathcal{X} = \{X_1, \ldots, X_n\}$ and a joint distribution $P$ that factorizes over $D$. The parents (children) of a variable $X_i$ on graph $D$, denoted by $\text{Pa}_D(X_i)$ (or $\text{Ch}_D(X_i)$), are the set of variables $X_j$ such that there is a directed edge from $X_j$ to $X_i$ (or from $X_i$ to $X_j$) on graph $D$. The set of ancestors (descendants) of a variable $X_i$, denoted by $\text{An}_D(X_i)$ (or $\text{De}_D(X_i)$), are the set of variables $X_j$ such that there is a path from $X_j$ to $X_i$ (or from $X_i$ to $X_j$) on $D$. Without loss of generality, we assume the domain set for every $X_i$ is $\text{Dom}(X_i) = [K] := \{1, \ldots, K\}$. For every $X_i$, we write the set of neighbors of $X_i$ in graph $D$ as $N_D(X_i)$ including variables $X_j$ such that there is an edge between $X_j$ and $X_i$ regardless of the direction. The maximum degree of an undirected graph $G$ is denoted by $d_{\max}(G)$. Throughout, we denote the true causal graph by $D$, use $V(\cdot)$ as the set of vertices of a graph and define $\text{skeleton}(\cdot)$ as the undirected graph obtained by replacing the arrows in the directed graph with undirected edges.

**Definition 1** (Directed Tree and Causal Tree). *A directed tree is a DAG whose underlying undirected graph is a tree and all its edges point away from the root. A causal tree is a causal model whose underlying causal graph is a directed tree.*

For a node $X_i$ on a directed or undirected tree $D$ and its neighbor $Y \in N_D(X_i)$, we write $B_D^{X_i:Y}$ as the set of nodes that can be reached from $Y$ through any path on the graph (regardless of the directions of edges on the path), when the edge between $X_i$ and $Y$ is cut out from $D$. Note that the neighbor $Y$ itself is always included in branch $B_D^{X_i:Y}$.

**Definition 2** (Central Node (Greenewald et al., 2019)). *A central node $v_c$ of an undirected tree $\mathcal{T}$ with respect to a distribution $q$ over the nodes is one for which $\max_{j \in N_{\mathcal{T}}(v_c)} q(B_{\mathcal{T}}^{v_c:X_j}) \leq 1/2$. At least one such $v_c$ is guaranteed to exist for any distribution $q$ (Jordan, 1869; Greenewald et al., 2019).*

Informally, a central node $v_c$ guarantees that the weight of every branch around $v_c$ cannot be larger than $1/2$ according to the distribution $q(\cdot)$.

**Definition 3** (Essential Graph). *The class of causal DAGs that encode the same set of conditional independences is called the Markov equivalence class. Denote the Markov equivalence class of a DAG $D$ by $[D]$. The essential graph of $D$, denoted by $\mathcal{E}(D)$, has the same skeleton as $D$, with directed edges $X_i \to X_j$ if such edge direction between $X_i$ and $X_j$ holds for all DAGs in $[D]$, and undirected edges otherwise.*

The chain components of $\mathcal{E}(D)$, denoted by $\text{CC}(\mathcal{E}(D))$, are the connected components of $\mathcal{E}(D)$ after removing all directed edges. Every chain component of $\mathcal{E}(D)$ is a chordal graph (Andersson et al., 1997). A DAG whose essential graph has a single chain component is called a *moral DAG* (Greenewald et al., 2019).

**Definition 4** (Causal Forest (Greenewald et al., 2019)). *A causal graph is said to be a causal forest if each of the undirected components of the essential graph are trees.*

Many widely used causal DAGs including causal trees and bipartite causal graphs are examples of causal forest (Greenewald et al., 2019). Bipartite graph applications can range from biological networks, biomedical networks, biomolecular networks to epidemiological networks (Burgos et al., 2008; Kontou et al., 2016; Pavlopoulos et al., 2018).

### 3.2 Causal Bandit Problems

In causal bandit problems, the action set consists of interventions (Lattimore et al., 2016). An intervention on node $X$ removes all edges from $\text{Pa}_D(X)$ to $X$ and results in a post-intervention

distribution denoted by $P(\{X\}^c|\mathrm{do}(X = x))$ over $\{X\}^c \triangleq \mathcal{X} \setminus \{X\}$. An empty intervention is represented by $\mathrm{do}()$. The reward variable $\mathbf{R}$ is real-valued and for simplicity, we assume the reward is only directly influenced by one of the variables in $\mathcal{X}$, which we call the *reward generating variable* denoted by $X_R$ [1]. The learner does not know the identity of $X_R$. Since there is only one $X_R$ in our setting, the optimal intervention must be contained in the set of single-node interventions as follows: $\mathcal{A} = \{\mathrm{do}(X = x) \mid X \in \mathcal{X}, x \in [K]\}$. Thus, we focus on above intervention set with $|\mathcal{A}| = nK$ throughout the paper.

We denote the expected reward for intervention $a = \mathrm{do}(X = x)$ by $\mu_a = \mathbb{E}[\mathbf{R}|a]$. Then $a^* = \mathrm{argmax}_{a \in \mathcal{A}} \mu_a$ is the optimal action and we assume that $\mu_a \in [0, 1]$ for all $a$. A random reward for $a = \mathrm{do}(X = x)$ is generated by $\mathbf{R}|_a = \mu_a + \varepsilon$, where $\varepsilon$ is 1-subGaussian. At every round $t$, the learner pulls $a_t = \mathrm{do}(X_t = x_t)$ based on the knowledge from previous rounds and observes a random reward $R_t$ and the values of all $X \in \mathcal{X}$. The objective of the learner is to minimize the cumulative regret $R_T = T\mu_{a^*} - \sum_{t=1}^{T} \mu_{a_t}$ without knowing the causal graph $D$.

We make the following assumptions below.

**Assumption 1.** *The following three causal assumptions hold:*

- Causal sufficiency: *for every pair of observed variables, all their common causes are also observed.*
- Causal Markov condition: *every node in causal graph $G$ is conditionally independent of its nondescendents, given its parents.*
- Causal faithfulness condition: *the set of independence relations derived from Causal Markov condition is the exact set of independence relations.*

Assumption 1 is commonly made in causal discovery literature (Peters et al., 2012; Hyttinen et al., 2013; Eberhardt, 2017; Greenewald et al., 2019). Equivalently speaking, there is no latent common causes and the correpondence between d-separations and conditional independences is one-to-one.

**Assumption 2** (Causal Effect Identifiability). *There exists an $\varepsilon > 0$, such that for any two variables $X_i \to X_j$ in graph $D$, we have $|P(X_j = x|\mathrm{do}(X_i = x')) - P(X_j = x)| > \varepsilon$ holds for some $x \in Dom(X_j), x' \in Dom(X_i)$.*

Assumption 2 is necessary. It states that if there is a direct causal relation between two variables on the graph, the causal effect strength cannot go arbitrarily small. A similar version of this assumption was also made by Greenewald et al. (2019)) in their causal graph learning algorithms. We show the necessity of this assumption in Section 6. Intuitively, without this assumption, any causal relation among variables cannot be determined through finite intervention samples and $\Omega(\sqrt{nKT})$ worst-case regret is the best one can hope for.

**Assumption 3** (Reward Identifiability). *We assume that for all $X \in An(X_R)$, there exists $x \in [K]$ such that $|\mathbb{E}[\mathbf{R}|\mathrm{do}()] - \mathbb{E}[\mathbf{R}|\mathrm{do}(X = x)]| \geq \Delta$, for some universal constant $\Delta > 0$.*

Lastly, we show that Assumption 3 is also necessary. It guarantees a difference on the expected reward between the observations and after an intervention on an ancestor of $X_R$. In Section 6, we prove that the worst-case regret is again lower bounded by $\Omega(\sqrt{nKT})$ without this assumption.

In the following sections, we describe our causal bandit algorithms that focus on intervention design to minimize regret. Like most intervention design works, our algorithm take the essential graph $\mathcal{E}(D)$ and observational probabilities over $\mathcal{X}$ (denoted by $P(\mathcal{X})$) as the input, which can be estimated from enough observational data [2] (He and Geng, 2008; Greenewald et al., 2019; Squires et al., 2020).

## 4  CN-UCB for trees and forests

We start with introducing our algorithm CN-UCB (Algorithm 1) when the causal graph $D$ is a directed tree. Before diving into details, we summarize our results as follows:

**Theorem 1** (Regret Guarantee for Algorithm 1). *Define $B = \max\left\{\frac{32}{\Delta^2} \log\left(\frac{8nK}{\delta}\right), \frac{2}{\varepsilon^2} \log\left(\frac{8n^2K^2}{\delta}\right)\right\}$ and $T_1 = KB(2 + d)\log_2 n$, where $0 < \delta < 1$.*

---

[1] For multiple reward generating variables cases, one can repeatedly run our proposed algorithms and recover each of them one by one. We discuss this setting in Section C

[2] Observational data is usually much more cheaper than interventional data (Greenewald et al., 2019).

*Suppose we run CN-UCB with $T \gg T_1$ interventions in total and $T_2 := T - T_1$. Then with probability at least $1 - \delta$, we have*

$$R_T = \widetilde{O}\left(K \max\left\{\frac{1}{\Delta^2}, \frac{1}{\varepsilon^2}\right\} d(\log n)^2 + \sqrt{KT}\right), \tag{1}$$

*where $d := d_{\max}(\text{skeleton}(D))$ and $\widetilde{O}$ ignores poly-log terms non-regarding to $n$.*

From above results, we see that especially for large $n$ and small maximum degree $d$, Algorithm 1 outperforms the standard MAB bandit approaches that incur $\widetilde{O}(\sqrt{nKT})$ regret.

## 4.1 Description for CN-UCB

Our method contains three main stages. There is no v-structure in a directed tree, so the essential graph obtained from observational data is just the tree skeleton $\mathcal{T}_0$ and we take it as our input in Algorithm 1. In stage 1, Algorithm 1 calls Algorithm 3 to find a directed sub-tree $\widetilde{\mathcal{T}}_0$ that contains $X_R$ by performing interventions on the central node (may change over time) on the tree skeleton. In stage 2, Algorithm 1 calls Algorithm 4 to find the reward generating node $X_R$ by central node interventions on $\widetilde{\mathcal{T}}_0$. We prove that $X_R$ can be identified correctly with high probability from the first two stages, so a UCB algorithm can then be applied on the reduced intervention set $\mathcal{A}_R = \{\text{do}(X_R = k) \mid k = 1, \ldots, K\}$ for the remaining rounds in stage 3. In the remainder of this section, we explain our algorithm design for each stage in detail and use an example in Figure 1 to show how CN-UCB proceeds step by step.

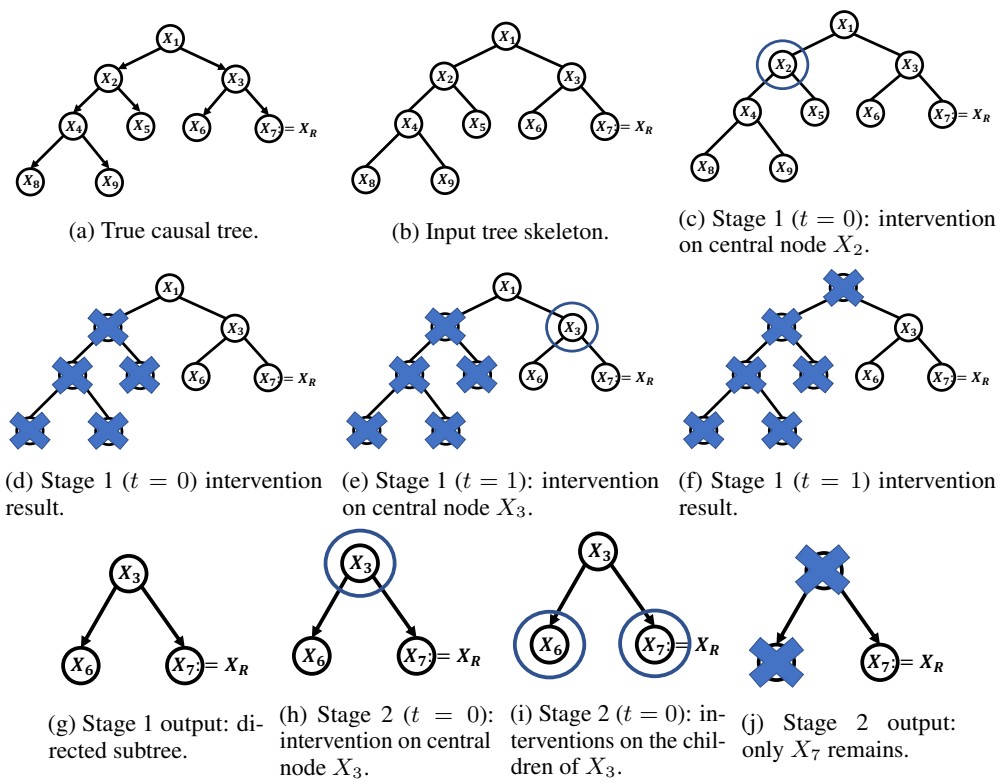

(a) True causal tree.

(b) Input tree skeleton.

(c) Stage 1 ($t = 0$): intervention on central node $X_2$.

(d) Stage 1 ($t = 0$) intervention result.

(e) Stage 1 ($t = 1$): intervention on central node $X_3$.

(f) Stage 1 ($t = 1$) intervention result.

(g) Stage 1 output: directed subtree.

(h) Stage 2 ($t = 0$): intervention on central node $X_3$.

(i) Stage 2 ($t = 0$): interventions on the children of $X_3$.

(j) Stage 2 output: only $X_7$ remains.

Figure 1: Causal Tree Example for CN-UCB procedure.

In the beginning of CN-UCB, we collect reward data under empty intervention to obtain the observational reward mean estimate $\hat{R}$, which will be used in later steps to check whether a variable is an ancestor of the true reward generating variable $X_R$ or not.

In our example, the true causal graph and its essential graph are displayed in Figure 1a and 1b, where the true reward generating variable is $X_7$.

---
**Algorithm 1** Central Node Upper Confidence Bound (CN-UCB)
---
1: **Input:** Tree skeleton $\mathcal{T}_0$, $K$, $\Delta$, $\varepsilon$, $B$, $T_2$, observational probabilities $P(\mathcal{X})$.
2: Perform do() for $B$ times, collect $R_1, \ldots, R_B$
3: Obtain reward estimate for the empty intervention do(): $\hat{R} \leftarrow \frac{1}{B} \sum_{b=1}^{B} R_b$.
4: **Stage 1:** Find a directed subtree that contains $X_R$: $\widetilde{\mathcal{T}}_0 \leftarrow$ Find Sub-tree($\mathcal{T}_0, K, B, \varepsilon, \Delta, \hat{R}$).
              //call Algorithm 3 in Section B.
5: **Stage 2:** Find the key node that generates the reward: $X_R \leftarrow$ Find Key Node($\widetilde{\mathcal{T}}_0, K, B, \Delta, \hat{R}$).
              //call Algorithm 4 in Section B.
6: **Stage 3:** Apply UCB algorithm on $\mathcal{A}_R = \{\mathrm{do}(X_R = k) \mid k = 1, \ldots, K\}$ for $T_2$ rounds.
---

**Stage 1.** The goal of this stage is to find a *directed* subtree that contains $X_R$ within $O(\log n)$ single-node interventions. We achieve this goal by adaptively intervening the central node that may change over time. To illustrate our main idea, we make the following claim for single-node interventions.

**Claim 1** (Two outcomes from single-node interventions on variable $X$). *1) Whether $X$ is an ancestor of $X_R$ or not can be determined. 2) The directed subtree induced by $X$ as its root can be found.*

To see the first outcome, we obtain reward estimates under interventions over $X$. If the difference between any of the reward estimates and $\hat{R}$ is larger than a threshold, Assumption 3 guarantees us that $X$ is an ancestor of $X_R$.

To see the second outcome, we estimate interventional probabilities $\hat{P}(Y = y | \mathrm{do}(X = x))$ for all $Y \in N_{\mathcal{T}_0}(X)$ and $y, x \in [K]$. Comparing these quantities with the corresponding observational probabilities $P(Y = y)$, the directions of edges between $X$ and its neighbors can be determined due to Assumption 2. Note that in a tree structure, at most one of the neighbors has causal effect on $X$ while others are all causally influenced by $X$. Once the edges between $X$ and its neighbors are oriented, all other edges in the subtree induced by $X$ (as the root) can be oriented using the property that each variable has at most one parent in a tree structure. Thus, if we learn that $X$ is indeed an ancestor of $X_R$, a directed subtree can be obtained immediately. Otherwise, we conclude that all the variables in the directed subtree induced by $X$ cannot be $X_R$.

Then a key question is: how do we adaptively select which variables to intervene? Arbitrarily selecting variables may easily require $O(n)$ single-node interventions. For example, let us consider a graph $X_1 \rightarrow X_2 \rightarrow \cdots \rightarrow X_n$ and the reward generating variable is $X_1$. If we start from intervening on $X_n$ and everytime move one-node backward towards intervening $X_1$, we need $O(n)$ single-node interventions in total to figure out the directed subtree containing $X_1$. To overcome this issue, we adopt the idea of central node algorithm proposed by Greenewald et al. (2019). Interventions on a central node $v_c(t)$ guarantees us to eliminate at least half of the nodes from future searching, because the directed subtree induced by $v_c(t)$ contains more than half of the nodes regarding to the current weighting function $q_t(\cdot)$. And as long as $v_c(t)$ is not found as an ancestor of $X_R$, the weights $q_{t+1}(\cdot)$ for variables in the directed subtree induced by $v_c(t)$ as the root will be set as zeros. Thus, stage 1 can finish within $O(\log n)$ single-node interventions on $\mathcal{T}_0$.

For the example in Figure 1, CN-UCB takes the tree skeleton in Figure 1b as its input and identifies $X_2$ as the central node. As Figure 1d shows, the directed subtree induced by root $X_2$ are crossed out from the candidate set for $X_R$ since intervening on $X_2$ shows that $X_2$ is not an ancestor of the true $X_R$. CN-UCB then identifies $X_3$ as the central node over the remaining variables(Figure 1e). We will see $X_3$ is indeed an ancestor of $X_7$ from interventions on $X_3$. Thus, $X_1$ can be crossed out because $X_1$ is on the upstream direction of $X_3$ and cannot be the true $X_R$ (Figure 1f). Finally, stage 1 outputs the directed subtree in Figure 1g whose root is $X_3$.

**Stage 2.** The goal of this stage is to identify $X_R$ in the directed sub-tree $\widetilde{\mathcal{T}}_0$ within $O(\log n)$ single-node interventions. Similar to stage 1, we determine whether a node $X$ is an ancestor of $X_R$ by performing interventions on it. If we find a variable $X$ that is not an ancestor of $X_R$ according to its reward estimates, all nodes in the sub-tree induced by $X$ can be eliminated from future searching. Otherwise, we continue to intervene on the children of $X$. Since $X_R$ is unique, at most one child $Y \in \mathrm{Ch}(X)$ is an ancestor of $X_R$. If such $Y$ exists, we repeat above procedure on the directed

sub-tree induced by $Y$ as the root and update weights for all other variables as zeros in the algorithm. Lastly, if none of the children $Y \in \mathrm{Ch}(X)$ appears to be an ancestor of $X_R$, $X$ itself must be $X_R$.

We again perform intervention on the central node at the beginning of each round in Algorithm 4 and intervene on its children if necessary. By the definition of central node, every round we can either eliminate at least half of the nodes or finish searching for $X_R$. Thus, stage 2 can be finished within $O(d \log_2 n)$ single node interventions on $\widetilde{\mathcal{T}}_0$, since the central node at each round has at most $d$ children.

For our example, stage 2 identifies $X_3$ as the central node showing in Figure 1h. With high probability we can conclude $X_3$ is indeed an ancestor of $X_7 := X_R$, so stage 2 continues to intervene on its children $X_6$ and $X_7$ (Figure 1i). From the intervention results we will see that $X_7$ is also an ancestor of the true $X_R$ so that $X_3$ and $X_6$ can be removed. Finally, stage 2 outputs $X_7$ with high probability (Figure 1j).

**Stage 3.** Once the first two stages output a variable, we simply apply the UCB algorithm over the reduced intervention set defined in Algorithm 1.

### 4.2 Extension of CN-UCB to causal forest

Generalizing CN-UCB to causal forest[3] defined in Definition 4 is straightforward. The causal forest version for CN-UCB is presented in Algorithm 5 (Section B). We simply run stage 1 and stage 2 of Algorithm 1 for every chain component (tree structure) of the causal forest essential graph until stage 2 finds the reward generating variable $X_R$ with high probability. Then, a standard UCB algorithm can be applied on the reduced intervention set corresponding to $X_R$. The regret guarantee for Algorithm 5 is presented in Theorem 2.

**Theorem 2** (Regret Guarantee for Algorithm 5). *Define $C(D)$ as the number of chain components in $\mathcal{E}(D)$. Suppose we run Algorithm 5 (Section 5) with $T \gg 2KB(d + C(D)) \log_2 n := T_1$ total interventions and $T_2 = T - T_1$, where $0 < \delta < 1$ and $B = \max\left\{ \frac{32}{\Delta^2} \log\left(\frac{8nK}{\delta}\right), \frac{2}{\varepsilon^2} \log\left(\frac{8n^2K^2}{\delta}\right) \right\}$. Then under Assumptions 1, 2 and 3, with probability at least $1 - \delta$, we have*

$$R_T = \widetilde{O}\left( K \max\left\{ \frac{1}{\Delta^2}, \frac{1}{\varepsilon^2} \right\} (d + C(D))(\log n)^2 + \sqrt{KT} \right), \tag{2}$$

*where $\widetilde{O}$ ignores poly-log terms non-regarding to $n$ and $d := \max_{\mathcal{T}_0 \in CC(\mathcal{E}(D))} d_{\max}(\mathcal{T}_0)$.*

## 5 Extension of CN-UCB to a general class of causal graphs

In this section, we extend CN-UCB to more general causal graphs. Our approach involves searching for the reward generating node $X_R$ within each undirected chain component of $\mathcal{E}(D)$. In order to use the central node idea, we try to construct tree structures on each component.

### 5.1 Preliminaries for the general graph class

Before describing our algorithm, we first review definitions of clique (junction) tree and clique graph for an undirected graph $G$ following Squires et al. (2020).

**Definition 5** (Clique). *A clique $C \subset V(G)$ is a subset of nodes in which every pair of nodes are connected with an edge. We say a clique $C$ is maximal if for any $v \in V(G) \setminus C$, $C \cup \{v\}$ is not a clique. The maximal cliques set of $G$ is denoted by $\mathcal{C}(G)$ with its clique number defined as $\omega(G) = \max_{C \in \mathcal{C}(G)} |C|$, where $|C|$ denotes the number of nodes in clique $C$. We use $|\mathcal{C}(G)|$ to denote the number of cliques in set $\mathcal{C}(G)$.*

**Definition 6** (Clique Tree (Junction Tree)). *A clique tree $\mathcal{T}_G$ for $G$ is a tree with maximal cliques $\mathcal{C}(G)$ as vertices that satisfies the junction tree property, i.e. for all $v \in V(G)$, the induced subgraph on the set of cliques containing $v$ is a tree. We denote the set of clique trees for graph $G$ by $\mathcal{T}(G)$.*

---

[3]Note that in our setting, causal forest refers to a type of causal graph. This should not be confused with the causal forest machine learning method which is similar to random forest.

---

**Algorithm 2** CN-UCB for General Causal Graph

---

1: **Input:** essential graph $\mathcal{E}(D), K, \Delta, \varepsilon, B, T_2$, observational probabilities $P(\mathcal{X})$.
2: **Initialize:** found $\leftarrow$ False
3: Perform do() for $B$ times, collect reward data $R_1, \ldots, R_B$; $\hat{R} \leftarrow \frac{1}{B} \sum_{b=1}^{B} R_b$.
4: **for** $G \in \mathrm{CC}(\mathcal{E}(D))$ **do**
5:     **Stage 1:** Create a junction tree $\mathcal{T}_G$ and find a directed sub-junction-tree that contains $X_R$:
    $\widetilde{\mathcal{T}}_G \leftarrow$ Find Sub-junction-tree$(G, \mathcal{T}_G, K, B, \varepsilon, \Delta, \hat{R})$.       //call Algorithm 7 in Section B.
6:     **if** $\widetilde{\mathcal{T}}_G$ is not empty **then**
7:         **Stage 2:** Find a clique in $\widetilde{\mathcal{T}}_G$ that contains $X_R$: $C_0 \leftarrow$ Find Key Clique$(\widetilde{\mathcal{T}}_G, K, B, \Delta, \hat{R})$.
                //call Algorithm 8 in Section B.
8:         **Stage 3:** Apply UCB algorithm on $\mathcal{A}_{C_0} = \{\mathrm{do}(X = k) \mid X \in V(C_0), k = 1, \ldots, K\}$ for $T_2$ rounds.
9:         **return** Finished.
10:     **end if**
11: **end for**
12: **return** No clique that contains $X_R$ is found. Apply UCB algorithm on the entire action space $\mathcal{A} = \{\mathrm{do}(X = x) \mid X \in \mathcal{X}, x \in [K]\}$.

---

**Definition 7** (Clique Graph). *A clique graph $\Gamma_G$ is the graph union of $\mathcal{T}(G)$, i.e. $V(\Gamma_G) = \mathcal{C}(G)$ and $U(\Gamma_G) = \cup_{\mathcal{T} \in \mathcal{T}(G)} U(\mathcal{T})$, where $U(\cdot)$ denotes the set of undirected edges.*

**Definition 8** (Directed Clique Tree). *A directed junction tree $\mathcal{T}_D$ of a moral DAG $D$ has the same vertices and adjacencies as a clique tree $\mathcal{T}_G$ of $G = \mathrm{skeleton}(D)$. We use asterisks as wildcards for edge points, e.g. $X_i * \to X_j$ denotes either $X_i \to X_j$ or $X_i \leftrightarrow X_j$. For each ordered pair of adjacent cliques $C_1 * - * C_2$ we orient the edge mark of $C_2$ as: 1) $C_1 * \to C_2$ if for $\forall v_{12} \in C_1 \cap C_2$ and $\forall v_2 \in C_2 \setminus C_1$, we have $v_{12} \to v_2$ in the DAG $D$ or 2) $C_1 * - C_2$ otherwise.*

**Definition 9** (Directed Clique Graph). *The directed clique graph $\Gamma_D$ of a moral DAG $D$ is the graph union of all directed clique trees of $D$. The procedure of graph union is the same as Definition 8.*

**Definition 10** (Intersection Comparable (Squires et al., 2020)). *A pair of edges $C_1 - C_2$ and $C_2 - C_3$ on a clique graph are intersection comparable if $C_1 \cap C_2 \subseteq C_2 \cap C_3$ or $C_1 \cap C_2 \supseteq C_2 \cap C_3$.*

Intersection-incomparable chordal graphs were introduced as "uniquely representable chordal graphs" in Kumar and Madhavan (2002). This class includes many familiar graph classes such as causal trees, causal forests and proper interval graphs (Squires et al., 2020). A clique graph is said to be intersection-incomparable if above intersection-comparable relation does not hold for any pair of edges. In this section, we design our algorithm for general causal graphs. If for every chain component $G \in \mathrm{CC}(\mathcal{E}(D))$, the corresponding clique graph $\Gamma_G$ is intersection-incomparable, we prove that our algorithm can achieve improved regret guarantee.

## 5.2 CN-UCB for general causal graphs

Our approach for causal bandit problems with general causal graphs is similar as previous settings. We try to find a subset of variables $\mathcal{X}_{\mathrm{sub}}$ that contain the true reward generating node $X_R$ with high probability and then apply UCB algorithm on interventions over $\mathcal{X}_{\mathrm{sub}}$.

Our method is presented in Algorithm 2. We search for $X_R$ in every chain component $G \in \mathrm{CC}(\mathcal{E}(D))$ separately. In stage 1, we create a junction tree $\mathcal{T}_G$ and search for a directed sub-junction-tree $\widetilde{\mathcal{T}}_G$ that contains $X_R$ via Algorithm 7. If such a $\widetilde{\mathcal{T}}_G$ exists, we keep searching for a clique $C_0$ in $\widetilde{\mathcal{T}}_G$ that contains $X_R$ in stage 2 via Algorithm 8 and apply UCB algorithm on single-node interventions over variables in $C_0$. Otherwise, we repeat above procedure for the next chain component.

Stage 1 and stage 2 in Algorithm 2 also use the idea of central node interventions. But different from previous settings, Algorithm 7 and Algorithm 8 proceed by performing interventions on variables in the central clique that changes over time and every central clique intervention can eliminate at least half of the cliques on $\mathcal{T}_G$ or $\widetilde{\mathcal{T}}_G$ from future searching. A central clique for a junction tree is defined in the same way as a central node for a tree in that we just treat the cliques on a junction tree as the "node" in Definition 2.

**Remark on the intersection-incomparable property.** We now explain why we need the intersection-incomparable property like other central node based algorithm that also involves constructing clique graphs (Squires et al., 2020). In general, it is possible for a directed junction tree $\mathcal{T}$ to have v-structures over cliques, which can make us unable to finish searching for a directed sub-junction-tree that contains $X_R$ by intervening on variables in $O(\log|\mathcal{T}|)$ cliques, where $|\mathcal{T}|$ denotes the number of cliques on $\mathcal{T}$. The reason is, a clique node may have more than one clique parents in $\mathcal{T}$, so that we cannot eliminate more than half of the cliques by interventions over the central clique. In total, we may need to perform $O(n)$ single-node interventions to find $X_R$ and incur $O(n)$ regret. However, for every chain component $G$ of the essential graph, if its clique graph $\Gamma_G$ is intersection-incomparable, then there is no v-structure over cliques in any junction tree $\mathcal{T}_G$ of $G$ (Squires et al., 2020), i.e. every node has at most one parent node.

Algorithm 2 can take any graph input no matter whether intersection-incomparable property holds or not. If the property does not hold, Algorithm 2 may output nothing at line 5 or line 7 for all component $G \in \text{CC}(\mathcal{E}(D))$ (will not output an incorrect sub-junction tree or clique by its design). Thus, if the learner finds that Algorithm 2 does not output anything after line 5 or 7 for all components, standard UCB algorithm can be used on the entire action set (see line 12 in Algorithm 2).

**Theorem 3** (Regret guarantee for Algorithm 2 ($\Gamma_G$ is intersection-incomparable for all $G \in \text{CC}(\mathcal{E}(D))$)). *Define $0 < \delta < 1$ and $B = \max\left\{\frac{32}{\Delta^2}\log\left(\frac{8nK}{\delta}\right), \frac{2}{\varepsilon^2}\log\left(\frac{8n^2K^2}{\delta}\right)\right\}$. Suppose we run Algorithm 2 for $T \gg B + KB\log_2 n\left(\omega(G_R) + d\omega(G_R) + \sum_{G\in CC(\mathcal{E}(D))}\omega(G)\right) := T_1$ total number of interventions and $T_2 := T - T_1$. Then under Assumptions 1, 2 and 3, with probability at least $1 - \delta$, we have*

$$R_T = $$
$$\tilde{O}\left(\left(d(\mathcal{T}_{G_R})\omega(G_R) + \sum_{G\in CC(\mathcal{E}(D))}\omega(G)\right)K\max\left\{\frac{1}{\Delta^2}, \frac{1}{\varepsilon^2}\right\}\log n\log\mathcal{C}_{max} + \sqrt{\omega(G_R)KT}\right)$$
$$(3)$$

*where $d(\mathcal{T}_{G_R})$ denotes the maximum degree of junction tree $\mathcal{T}_{G_R}$, $G_R$ denotes the chain component that contains the true $X_R$ and $\mathcal{C}_{max} \triangleq \max_{G\in CC(\mathcal{E}(D))}|\mathcal{C}(G)|$. $\tilde{O}(\cdot)$ ignores poly-log terms non-regarding to $n$ or number of cliques.*

Above regret bound shows that our algorithm outperforms standard MAB algorithms especially when $n$ is large and the degree $d(\mathcal{T}_{G_R})$ and the clique numbers $\omega(G)$ are small . Also, above result reduces to Theorem 2 when $D$ is a causal forest, because $\omega(G)$ is always 2 for tree-structure chain components.

# 6  Lower bounds

In this section we show without Assumption 2 or 3, any algorithm will incur an $\Omega\left(\sqrt{nkT}\right)$ regret in the worst-case, which is exponentially worse than our results in terms of $n$.

**Definition 11** ($nK$-arm Gaussian Causal Bandit Class). *For a bandit instance in $nK$-arm Gaussian causal bandit class, actions are composed by single-node interventions over $n$ variables $\mathcal{X} = \{X_1, \ldots, X_n\}$: $\mathcal{A} = \{\text{do}(X_i = x)|x \in [K]; i = 1, \ldots, n\}$ with $|\mathcal{A}| = nK$. The reward for every action is Gaussian distributed and is directly influenced by one of $\mathcal{X}$.*

**Theorem 4** (Minimax Lower Bound Without Assumption 2). *Let $\mathcal{E}$ be the set of $nK$-armed Gaussian bandits (Definition 11) where the instances in $\mathcal{E}$ does not satisfy Assumption 2. We show that the minimax regret is $\inf_{\pi\in\Pi}\sup_{\nu\in\mathcal{E}}\mathbb{E}[R_T(\pi, \nu)] = \Omega(\sqrt{nKT})$, where $\Pi$ denotes the set of all policies.*

**Theorem 5** (Minimax Lower Bound (Assumption 2 holds but Assumption 3 does not hold)). *Let $\mathcal{E}$ be the set of $nK$-armed Gaussian bandits (Definition 11) where the instances in $\mathcal{E}$ satisfy Assumption 2 but does not satisfy Assumption 3. We show that the minimax regret is $\inf_{\pi\in\Pi}\sup_{\nu\in\mathcal{E}}\mathbb{E}[R_T(\pi, \nu)] = \Omega(\sqrt{nKT})$, where $\Pi$ denotes the set of all policies.*

# 7 Discussion

In this paper, we studied causal bandit problems with unknown causal graph structures. We proposed an efficient algorithm CN-UCB for the causal tree setting. The regret of CN-UCB scales logarithmically with the number of variables $n$ when the intervention size is $nK$. CN-UCB was then extended to broader settings where the underlying causal graph is a causal forest or belongs to a general class of graphs. For the later two settings, our regret again only scales logarithmically with $n$. Lastly, we provide lower bound results to justify the necessity of our assumptions on the causal effect identifiability and the reward identifiability. Our approaches are the first set of causal bandit algorithms that do not rely on knowing the causal graph in advance and can still outperform standard MAB algorithms.

There are several directions for future work. First, one can generalize CN-UCB to the multiple reward generating node setting, i.e., more than one variable directly influences the reward. We expect this can be done by repeatedly applying stage 1 and 2 in CN-UCB to find all the reward generating nodes one by one and apply a standard MAB algorithm on the reduced intervention set. In this setting, it is natural to consider interventions that can be performed on more than one variables. Second, it will be interesting to develop instance dependent regret bounds, for example, through estimating the probabilities over the causal graph in CN-UCB. Lastly, one can also develop efficient algorithms that do not need to know the causal graph when confounders exist.

## Acknowledgement

We thank our NeurIPS reviewers and meta-reviewer for helpful suggestions to improve the paper.

## Funding transparency statement

**Funding (financial activities supporting the submitted work):** Funding in direct support of this work: NSF CAREER grant IIS-1452099, Adobe Data Science Research Award.

**Competing Interests (financial activities outside the submitted work):** None

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
