# A  Proof for Theorems

## A.1  Proof for Theorem 1

We first present a sample complexity result for interventions in stage 1 and 2.

**Lemma 1** (Sample Complexity for Algorithm 1 Before Stage 3)**.** *Set* $B = \max\{\frac{32}{\Delta^2} \log\left(\frac{8nK}{\delta}\right), \frac{2}{\varepsilon^2} \log\left(\frac{8n^2K^2}{\delta}\right)\}$, *with probability at least* $1 - \delta$, *stage 2 in Algorithm 1 outputs the true key node within* $KB(2 + d) \log_2 n + B$ *interventions, where* $d$ *is defined as the maximum degree of the causal tree skeleton.*

*Proof for Lemma 1.* By Hoeffding's inequality for bounded random variables, for any fixed $X, Y \in \mathcal{X}, x \in [K], y \in [K]$, we have

$$|P(Y = y|\text{do}(X = x)) - \hat{P}(Y = y|\text{do}(X = x))| \leq \sqrt{\frac{1}{2B} \log\left(\frac{8n^2K^2}{\delta}\right)},$$

with probability at least $1 - \frac{\delta}{4n^2K^2}$. Furthermore, for fixed $X \in \mathcal{X}$ and $x \in [K]$, by Hoeffding's inequality for sub-gaussian random variables, we have

$$|\hat{R}^{\text{do}(X=x)} - \mathbb{E}[\mathbf{R}|\text{do}(X = x)]| \leq \sqrt{\frac{2}{B} \log\left(\frac{8nK}{\delta}\right)},$$

with probability at least $1 - \frac{\delta}{4nK}$. For empty intervention, we write $\hat{R}$ for $\hat{R}^{\text{do}()}$, then

$$|\hat{R} - \mathbb{E}[\mathbf{R}|\text{do}()]| \leq \sqrt{\frac{2}{B} \log\left(\frac{4}{\delta}\right)}$$

holds with probability at least $1 - \frac{\delta}{2}$. We define the union of above good events by $E$. By union bound, we have $P(E^c) \leq \delta$.

Under event $E$, suppose $X$ is the key node, one can show that

$$|\hat{R} - \hat{R}^{\text{do}(X=x)}| \geq$$

$$|\mathbb{E}[\mathbf{R}|\text{do}()] - \mathbb{E}[\mathbf{R}|\text{do}(X = x)]| - |\hat{R} - \mathbb{E}[\mathbf{R}|\text{do}()]| - |\mathbb{E}[\mathbf{R}|\text{do}(X = x)] - \hat{R}^{\text{do}(X=x)}| \geq \frac{\Delta}{2}.$$

And for any $X, Y, x, y$ such that $X \to Y$,

$$|P(Y = y) - \hat{P}(Y = y|\text{do}(v_c(t) = z))|$$

$$\geq |P(Y = y) - P(Y = y|\text{do}(v_c(t) = z))| - |P(Y = y|\text{do}(v_c(t) = z)) - \hat{P}(Y = y|\text{do}(v_c(t) = z))|$$

$$\geq \frac{\varepsilon}{2}.$$

Combine everything before stage 3, the total number of interventions is $2KB \log_2 n + dKB \log_2 n + B$. □

*Proof for Theorem 1.* Condition on the good event E in the proof of Lemma 1, stage 2 in Algorithm 1 returns the true reward generating node $X_R$. Combining with the total intervention sample results in Lemma 1 we have

$$R_T \leq KB(2 + d) \log_2 n + B + \sqrt{KT \log T}$$

$$= O\left(K \max\left\{\frac{1}{\Delta^2}, \frac{1}{\varepsilon^2}\right\} d \log\left(\frac{nK}{\delta}\right) \log_2 n + \sqrt{KT \log T}\right).$$

holds with probability at least $1 - \delta$. □

## A.2 Proof for Theorem 2

*Proof.* The proof is straightforward from Theorem 1. In Algorithm 5, there is only one tree component that contains $X_R$. Thus, in the worst case, we find $X_R$ in the last component and stage 1 needs to be performed for $C(D)$ times. Combine with the results in Section A.1, the total number of interventions is $C(D)KB\log_2 n + KB\log_2 n + dKB\log_2 n + B$. Then, the regret can be bounded by:

$$R_T \leq KB(d + C(D) + 1)\log_2 n + B + \sqrt{KT\log T}$$
$$= O\left(K\max\left\{\frac{1}{\Delta^2}, \frac{1}{\varepsilon^2}\right\}(d + C(D))\log\left(\frac{nK}{\delta}\right)\log_2 n + \sqrt{KT\log T}\right).$$

□

## A.3 Proof for Theorem 3

*Proof.* We follow the proof idea in Theorem 1.
By Hoeffding's inequality for bounded random variables, for any fixed $X, Y \in \mathcal{X}, x \in [K], y \in [K]$, we have

$$|P(Y = y|\mathrm{do}(X = x)) - \hat{P}(Y = y|\mathrm{do}(X = x))| \leq \sqrt{\frac{1}{2B}\log\left(\frac{8n^2K^2}{\delta}\right)},$$

with probability at least $1 - \frac{\delta}{4n^2K^2}$. Furthermore, for fixed $X \in \mathcal{X}$ and $x \in [K]$, by Hoeffding's inequality for sub-gaussian random variables, we have

$$|\hat{R}^{\mathrm{do}(X=x)} - \mathbb{E}[\mathbf{R}|\mathrm{do}(X = x)]| \leq \sqrt{\frac{2}{B}\log\left(\frac{8nK}{\delta}\right)},$$

with probability at least $1 - \frac{\delta}{4nK}$. For empty intervention, we write $\hat{R}$ for $\hat{R}^{\mathrm{do}()}$, then

$$|\hat{R} - \mathbb{E}[\mathbf{R}|\mathrm{do}()]| \leq \sqrt{\frac{2}{B}\log\left(\frac{4}{\delta}\right)}$$

holds with probability at least $1 - \frac{\delta}{2}$. We define the union of above good events by $E$. By union bound, we have $P(E^c) \leq \delta$. We condition on event $E$ for the following analysis.

By the design of Algorithm 2, we search for the reward generating node in every chain component of $\mathrm{CC}(\mathcal{E}(D))$. In the worst case, we find $X_R$ in the last component. In this case, Algorithm 7 (stage 1) is executed for every component $G$. and Algorithm 8 (stage 2) is executed only for the chain component that contains $X_R$. We use $|\mathcal{C}(G)|$ to denote the number of maximal cliques of component $G$ for every $G \in \mathrm{CC}(\mathcal{E}(D))$.

Specifically, for every chain component $G$, Algorithm 7 performs $KB\log_2 |\mathcal{C}(G)|$ clique interventions, i.e. at most $KB\omega(G)\log_2 |\mathcal{C}(G)|$ node interventions. For component $G_R$ that contains $X_R$, Algorithm 8 performs $(KB + d(\mathcal{T}_{G_R})KB)\log_2 |\mathcal{C}(G_R)|$ clique interventions, i.e. at most $(KB + d(\mathcal{T}_{G_R})KB)\omega(G_R)\log_2 |\mathcal{C}(G_R)|$ node interventions.

Thus, the total number of interventions before applying UCB algorithm on the reduced intervention set is: $\sum_{G \in \mathrm{CC}(\mathcal{E}(G))} KB\omega(G)\log_2 |\mathcal{C}(G)| + dKB\omega(G_R)\log_2 |\mathcal{C}(G_R)|$. Thus, conditioning on event $E$ which holds with at least $1 - \delta$, we have

$$R_T \leq \sum_{G \in \mathrm{CC}(\mathcal{E}(D))} KB\omega(G)\log_2 |\mathcal{C}(G)| + d(\mathcal{T}_{G_R})KB\omega(G_R)\log_2 |\mathcal{C}(G_R)| + \sqrt{\omega(G_R)KT\log T}$$
$$= \widetilde{O}\Big(K\max\{\frac{1}{\Delta^2}, \frac{1}{\varepsilon^2}\}\log n(d(\mathcal{T}_{G_R})\omega(G_R)\log_2 |\mathcal{C}(G_R)| + \sum_{G \in \mathrm{CC}(\mathcal{E}(D))} \omega(G)\log_2 |\mathcal{C}(G)|)$$
$$+ \sqrt{\omega(G_R)KT}\Big)$$

where $\widetilde{O}(\cdot)$ ignores poly-log terms non-regarding to $n$ or clique sizes.

□

## A.4 Proof for Theorem 4

*Proof.* We use an example to show that, without Assumption 2, no algorithm can achieve worst-case regret better than $\widetilde{O}(\sqrt{nKT})$.

Consider $(K+1)$-nary variables $X_1, \ldots, X_n \in \{0, 1, \ldots, K\}$ and action set $\mathcal{A} = \{\text{do}(X_i = x) | x \in \{1, \ldots, K\}; i = 1, \ldots, n\}$. Note that $|\mathcal{A}| = nK$.

We construct below $nK + 1$ bandit instances such that Assumption 2 does not hold.

**Bandit Instance** $0$:

- Causal structure $X_1 \to X_2 \to \cdots \to X_n$.

- Probabilities assigned: $\mathbb{P}(X_1 = 0) = \mathbb{P}(X_{i+1} = 0 | X_i) = 1, i = 1, \ldots, n-1$.[4]

- Reward generation: $R \sim N(0, 1)$.

- Note: for all $nK$ action, conditional reward mean is 0.

**Bandit Instance** $k$, **where** $k = 1, \ldots, K$:

- Causal structure $X_1 \to X_2 \to \cdots \to X_n$.

- Probabilities assigned: $\mathbb{P}(X_1 = 0) = \mathbb{P}(X_{i+1} = 0 | X_i) = 1, i = 1, \ldots, n-1$.

- Reward generation: $R \sim N(\Delta \mathbb{1}_{\{X_1 = k\}}, 1)$.

- Note: only action $\text{do}(X_1 = k)$ has reward mean $\Delta$, otherwise 0.

**Bandit Instance** $(j-1)K + k \sim jK$, **where** $j = 2, \ldots, n-1, k = 1, \ldots, K$:

- Causal structure $X_j \to X_1 \to \cdots \to X_{j-1} \to X_{j+1} \to \cdot \to X_n$.

- Probabilities assigned: $\mathbb{P}(X_j = 0) = \mathbb{P}(X_1 = 0 | X_j) = \mathbb{P}(X_{j+1} = 0 | X_{j-1}) = \mathbb{P}(X_{i+1} = 0 | X_i) = 1, i = 1, \ldots, j-2, j+1, \ldots, n-1$.

- Reward generation: $R \sim N(\Delta \mathbb{1}_{\{X_j = k\}}, 1)$.

- Note: only action $\text{do}(X_j = k)$ has reward mean $\Delta$, otherwise 0.

**Bandit Instance** $(n-1)K + k \sim nK$, **where** $k = 1, \ldots, K$:

- Causal structure $X_n \to \cdots \to X_2 \to X_1$.

- Probabilities assigned: $\mathbb{P}(X_n = 0) = \mathbb{P}(X_{i-1} = 0 | X_i) = 1, i = 2, \ldots, n$.

- Reward generation: $R \sim N(\Delta \mathbb{1}_{\{X_n = k\}}, 1)$.

- Note: only action $\text{do}(X_n = k)$ has reward mean $\Delta$, otherwise 0.

Take $\Delta = \frac{1}{4}\sqrt{\frac{nK}{T}}$, using the results in exercise 15.2 in Lattimore and Szepesvári (2018) we know that there exists one bandit intance $\nu$ in above such that

$$\mathbb{E}[R_T] \geq \frac{1}{8}\sqrt{nKT}, \tag{4}$$

where the expectation is taken over the entire randomness.

We present below lemmas in Lattimore and Szepesvári (2018) to describe the proof for above conclusion.

---

[4] $\mathbb{P}(X_{i+1} = 0 | X_i) = 1$ means no matter what value $X_i$ is, the conditional probability $\mathbb{P}(X_{i+1} = 0 | X_i)$ is always 1. Similar expression is also used for constructing other bandit instances.

**Lemma 2** (Divergence decomposition). *Let $\nu = (P_1, \ldots, P_k)$ be the reward distributions associated with one $k$-armed bandit, and let $\nu' = (P'_1, \ldots, P'_k)$ be the reward distributions associated with another $k$-armed bandit. Fix some policy $\pi$ and let $\mathbb{P}_\nu = \mathbb{P}_{\nu\pi}$ and $\mathbb{P}_{\nu'} = \mathbb{P}_{\nu'\pi}$ be the probability measures on the bandit model induced by the n-round interconnection of $\pi$ and $\nu$ ($\pi'$ and $\nu'$). Then*

$$\mathbf{KL}(\mathbb{P}_\nu, \mathbb{P}_{\nu'}) = \sum_{i=1}^{k} \mathbb{E}_\nu \left[ T_i(n) \right] \mathbf{KL}(P_i, P'_i). \tag{5}$$

**Lemma 3** (Pinsker inequality). *For measures $P$ and $Q$ on the same probability space $(\Omega, \mathcal{F})$ that*

$$\delta(P, Q) \triangleq \sup_{A \in \mathcal{F}} P(A) - Q(A) \leq \sqrt{\frac{1}{2} \mathbf{KL}(P, Q)}. \tag{6}$$

**Lemma 4.** *Let $(\Omega, \mathcal{F})$ be a measurable space and let $P, Q : \mathcal{F} \to [0, 1]$ be probability measures. Let $a < b$ and $X : \Omega \to [a, b]$ be a $\mathcal{F}$-measurable random variable, we have*

$$\left| \int_\Omega X(\omega) dP(\omega) - \int_\Omega X(\omega) dQ(\omega) \right| \leq (b - a) \delta(P, Q), \tag{7}$$

*where $\delta(P, Q)$ is as defined in Lemma 3.*

Define $\mu^{(l)} = \{\mu^{(l)}_{\text{do}(X_1=1)}, \ldots, \mu^{(l)}_{\text{do}(X_1=K)}, \ldots, \mu^{(l)}_{\text{do}(X_n=1)}, \ldots \mu^{(l)}_{\text{do}(X_n=K)}\} \in \mathbb{R}^{nK}$ as the reward mean vector for the $l$-th bandit instance in above. By construction, only the $l$-th element of $\mu^{(l)}$ is $\Delta$, and all other elements are zeros. For any policy $\pi$, we define $\mathbb{P}_l$ and $\mathbb{E}_l$ as the probability measure and expectation induced by bandit instance $l$ within $T$ rounds.

Denote the number of plays on arm $i$ up to time $T$ by $T_i(T)$, we have

$$\mathbb{E}_i \left[ T_i(T) \right] \leq \mathbb{E}_0 \left[ T_i(T) \right] + T \delta(\mathbb{P}_0, \mathbb{P}_i) \text{ by Lemma 4}$$

$$\leq \mathbb{E}_0 \left[ T_i(T) \right] + T \sqrt{\frac{1}{2} \mathbf{KL}(\mathbb{P}_0, \mathbb{P}_i)} \text{ by Lemma 3}$$

$$= \mathbb{E}_0 \left[ T_i(T) \right] + T \sqrt{\frac{1}{2} \cdot \frac{1}{2} \Delta^2 \mathbb{E}_0 \left[ T_i(T) \right]} \text{ by Lemma 2}$$

$$= \mathbb{E}_0 \left[ T_i(T) \right] + \frac{1}{8} \sqrt{nKT \mathbb{E}_0 \left[ T_i(T) \right]} \text{ by } \Delta = \frac{1}{4} \sqrt{\frac{nK}{T}}.$$

Sum over the left term in above, using the property $\sum_{i=1}^{nK} \mathbb{E}_0 \left[ T_i(T) \right] = T$, we have

$$\sum_{i=1}^{nK} \mathbb{E}_i \left[ T_i(T) \right] \leq T + \frac{\sqrt{nKT}}{8} \sum_{i=1}^{nK} \sqrt{\mathbb{E}_0 \left[ T_i(T) \right]}$$

$$\leq T + \frac{1}{8} nKT.$$

Let $R_i \triangleq R_T(\pi, \nu_i)$ (the regret of applying policy $\pi$ on the $i-$th bandit instance up to time $T$), where $\nu_i$ refers to the $i$-th bandit instance in above construction.

$$\sum_{i=1}^{nK} \mathbb{E}[R_i] = \Delta \sum_{i=1}^{nK} (T - \mathbb{E}_i \left[ T_i(T) \right])$$

$$\geq \frac{1}{4} \sqrt{\frac{nK}{T}} (nKT - T - \frac{1}{8} nKT) \geq \frac{nKT}{8} \sqrt{\frac{nK}{T}} = \frac{nK}{8} \sqrt{nKT}.$$

Thus, there exists one bandit instance $i^*$, such that $\mathbb{E}[R_{i^*}] \geq \frac{1}{8} \sqrt{nKT}$. $\qquad \square$

## A.5 Proof for Theorem 5

*Proof.* Consider $(K + 2)$-nary variables $X_1, \ldots, X_n \in \{0, 1, \ldots, K + 1\}$ and action set $\mathcal{A} = \{\text{do}(X_i = x) \mid x \in \{2, \ldots, K + 1\}; i = 1, \ldots, n\}$. Note that $|\mathcal{A}| = nK$.

We construct below $nK + 1$ bandit instances such that Assumption 2 holds but Assumption 3 does not hold.

**Bandit Instance** 0**:**

- Causal structure $X_n \to X_{n-1} \to \cdots \to X_1$.

- Probabilities assigned: $\mathbb{P}(X_n = 0) = 1$; $\mathbb{P}(X_{i-1} = 0 | X_i = 0) = \mathbb{P}(X_{i-1} = 1 | X_i = 1) = 1$; $P(X_{i-1} \geq 2 | X_i) = 0$ for any value of $X_i$. ($i = 2, \ldots, n$)

- Reward generation: $R \sim N(0, 1)$.

- Note: for all $nK$ action, their reward mean is 0.

**Bandit Instance** $k$**, where** $k = 1, \ldots, K$**:**

- Causal structure $X_n \to X_{n-1} \to \cdots \to X_1$.

- Probabilities assigned: $\mathbb{P}(X_n = 0) = 1$; $\mathbb{P}(X_{i-1} = 0 | X_i = 0) = \mathbb{P}(X_{i-1} = 1 | X_i = 1) = 1$; $P(X_{i-1} \geq 2 | X_i) = 0$ for any value of $X_i$. ($i = 2, \ldots, n$)

- Reward generation: $R \sim N(\Delta \mathbb{1}_{\{X_1 = k+1\}}, 1)$.

- Note: only action do$(X_1 = k + 1)$ has reward mean $\Delta$, otherwise 0.

**Bandit Instance** $(j-1)K + k \sim jK$**, where** $j = 2, \ldots, n-1, k = 1, \ldots, K$**:**

- Causal structure $X_{j-1} \to \cdots \to X_1 \to X_n \to X_{n-1} \to \cdots \to X_j$.

- Probabilities assigned: $\mathbb{P}(X_{j-1} = 0) = 1$, for other directly connected variable pairs $X \to Y$ in the graph we have $\mathbb{P}(Y = 0 | X = 0) = \mathbb{P}(Y = 1 | X = 1) = 1$ and $\mathbb{P}(Y \geq 2 | X) = 0$ for any value of $X$.

- Reward generation: $R \sim N(\Delta \mathbb{1}_{\{X_j = k+1\}}, 1)$.

- Note: only action do$(X_j = k + 1)$ has reward mean $\Delta$, otherwise 0.

**Bandit Instance** $(n-1)K + k \sim nK$**, where** $k = 1, \ldots, K$**:**

- Causal structure $X_{n-1} \to \cdots \to X_1 \to X_n$.

- Probabilities assigned: $\mathbb{P}(X_{n-1} = 0) = 1$, for other directly connected variable pairs $X \to Y$ in the graph we have $\mathbb{P}(Y = 0 | X = 0) = \mathbb{P}(Y = 1 | X = 1) = 1$ and $\mathbb{P}(Y \geq 2 | X) = 0$ for any value of $X$.

- Reward generation: $R \sim N(\Delta \mathbb{1}_{\{X_n = k+1\}}, 1)$.

- Note: only action do$(X_n = k + 1)$ has reward mean $\Delta$, otherwise 0.

In above $nK + 1$ instances, we see that Assumption 2 holds since $P(Y = 1) = 0$ but $P(Y = 1 | X = 1) = 1$ for all directly connected variable pairs $X \to Y$. However, Assumption 3 does not hold. For example in bandit instance 1, no matter how we intervene on $X_n, \ldots, X_2$, by probabilities construction $X_1 \geq 2$ can never happen, which means the expected reward is always zero unless we direcly intervene on $X_1$. Similarly, Assumption 3 does not hold for other bandit instances as well.

Follow the exact proof for Theorem 4, we know that there exists one bandit instance $\nu$ in above such that

$$\mathbb{E}[R_T(\pi, \nu)] = \Omega(\sqrt{nKT}), \tag{8}$$

for any policy $\pi$. $\qquad\square$

## B   Algorithms

We present our algorithms in this section.

---

**Algorithm 3** Find Sub-tree

---

1: **Input:** tree skeleton $\mathcal{T}_0$, $K$, $B$, $\varepsilon$, $\Delta$, $\hat{R}$.
2: **Initialize:** $t \leftarrow 0, q_0(X) \leftarrow 1/n$ for all $X \in \mathcal{X}$, found $\leftarrow$ False (indicating whether an ancestor of $X_R$ is found or not).
3: **while** found is False **do**
4:     Identify central node $v_c(t) \leftarrow$ Find Central Node$(\mathcal{T}_0, q_t(\cdot))$.        //Algorithm 6 in Section B.
5:     Initialize the edge direction between $Y$ and $v_c(t)$ as $Y \rightarrow v_c(t)$ by:
        direction$(Y) \leftarrow$ up, for all $Y \in N_{\mathcal{T}_0}(v_c(t))$.
6:     **for** $z \in [K]$ **do**
7:         Perform $a = \text{do}(v_c(t) = z)$ for $B$ times,
            collect interventional samples $Y_1^a, \ldots, Y_B^a$ for $Y \in N_{\mathcal{T}_0}(v_c(t))$ and $R_1^a, \ldots, R_B^a$.
8:         Estimate interventional probabilities $\hat{P}(Y = y \mid a) \leftarrow \frac{1}{B} \sum_{b=1}^{B} \mathbb{1}_{\{Y_b^a = y\}}$ for $y \in [K]$.
9:         **if** $|\hat{R} - \frac{1}{B} \sum_{b=1}^{B} R_b^a| > \Delta/2$ **then**
10:            found $\leftarrow$ True.                          //by Assumption 3, $v_c(t)$ is an ancestor of $X_R$.
11:        **end if**
12:        **for** $Y \in N_{\mathcal{T}_0}(v_c(t))$ **do**
13:            **if** $|P(Y = y) - \hat{P}(Y = y \mid a)| > \varepsilon/2$ for some $y \in [K]$ **then**
14:                Update the edge direction between $Y$ and $v_c(t)$ as $v_c(t) \rightarrow Y$ by:
                    direction$(Y) \leftarrow$ down.            //by Assumption 2, the edge direction is updated as
                    $v_c(t) \rightarrow Y$.
15:            **end if**
16:        **end for**
17:    **end for**
18:    **if** found is True **then**
19:        **return** $\widetilde{\mathcal{T}_0}$ induced by root $v_c(t)$.        //cut off upstream branches using direction$(\cdot)$ results.
20:    **end if**
21:    **for** $Y \in N_{\mathcal{T}_0}(v_c(t))$ **do**
22:        **if** direction$(Y) =$ down **then**
23:            $q_{t+1}(X) \leftarrow 0$, for $X \in B_{\mathcal{T}_0}^{v_c(t):Y} \cup \{v_c(t)\}$. //variables that cannot be an ancestor of $X_R$.
24:        **else**
25:            $q_{t+1}(X) \leftarrow 1$, for $X \in B_{\mathcal{T}_0}^{v_c(t):Y}$.                //variables that might be an ancestor of $X_R$.
26:        **end if**
27:    **end for**
28:    normalize $q_{t+1}(\cdot)$, $t \leftarrow t + 1$.
29: **end while**

---

## C   Discussion on Multiple Reward Generating Variables

In this section, we discuss how to generalize Algorithm 1 to the setting where multiple reward generating variables exist.

In this setting, we will have to generalize Assumption 3 to hold on interventions on any ancestor of each of the direct causes. That is to say for any variable $X$ that is an ancestor of certain direct cause of the reward, we have $|\mathbb{E}[R \mid \text{do}(X = x) - \mathbb{E}[R \mid \text{do}()]]| > \Delta$ for some $x$ in the domain of $X$. Then our approach can be generalized to this setting by running multiple times. Specifically, in Algorithm 1, we don't stop after stage 2 finds a reward generating variable, say $X_{R_1}$ because it is only one of the direct causes. By construction of stage 2 (Algorithm 4), we know except for $X_{R_1}$ itself, none of its descendants is a direct cause of the reward (we always check children first). We can fix the values of all variables in the subtree $\mathcal{T}_1$ induced by $X_{R_1}$ as the root. Then we re-run Algorithm 1 on the remaining graph (still a tree): original tree cut by subtree $\mathcal{T}_1$, and find a second direct cause of the reward. This procedure can be repeated until no further reward-generating variable can be found.

Above idea is a way to find multiple reward generating variables, but it's also possible to come up with more efficient ways. For example, instead of finding the direct causes one by one, it will be interesting to develop methods that can find several of them simultaneously. We think this setting itself is also interesting and worth studying as an independent work.

---

**Algorithm 4** Find Key Node

---

1: **Input:** directed sub-tree $\widetilde{\mathcal{T}}_0$, $K$, $B$, $\Delta$, $\hat{R}$.

2: **Initialize:** $t \leftarrow 0, q_0(X) \leftarrow 1/|\widetilde{\mathcal{T}}_0|$ for all $X \in V(\mathcal{T}_0)$.

3: **while** $q_t(X) > 0$ for more than one $X \in V(\widetilde{\mathcal{T}}_0)$ **do**

4:      Identify central node $v_c(t) \leftarrow$ Find Central Node $(\widetilde{\mathcal{T}}_0, q_t(\cdot))$.     //Algorithm 6 in Section B.

5:      Initialize direction $\leftarrow$ up. //Indicating whether the true $X_R$ is towards the upstream direction of $v_c(t)$ or downstream direction of $v_c(t)$.

6:      **for** $z \in [K]$ **do**

7:          Perform $a = \mathrm{do}(v_c(t) = z)$ for $B$ times, collect samples $R_1^a, \ldots, R_B^a$.

8:          **if** $|\hat{R} - \frac{1}{B}\sum_{b=1}^B R_b^a| > \Delta/2$ **then**

9:              direction $\leftarrow$ down.                               //by Assumption 3.

10:          **end if**

11:      **end for**

12:      **if** direction $=$ up **then**

13:          $q_{t+1}(X) \leftarrow 1$ for $X \in B_{\widetilde{\mathcal{T}}_0}^{v_c(t):\mathrm{Pa}_{\widetilde{\mathcal{T}}_0}(v_c(t))}$.     //variables that might be an ancestor of $X_R$.

            $q_{t+1}(v_c(t)) \leftarrow 0$.                   //$v_c(t)$ cannot be an ancestor of $X_R$ if direction is up.

14:          **for** $Y \in \mathrm{Ch}_{\widetilde{\mathcal{T}}_0}(v_c(t))$ **do**

15:              $q_{t+1}(X) \leftarrow 0$, for $X \in B_{\widetilde{\mathcal{T}}_0}^{v_c(t):Y}$.     //variables that cannot be an ancestor of $X_R$.

16:          **end for**

17:      **else**

18:          Initialize RewardBranch $\leftarrow$ None. //indicating the branch pointing from $v_c(t)$ that contains the true $X_R$.

19:          **for** $Y \in \mathrm{Ch}_{\widetilde{\mathcal{T}}_0}(v_c(t))$ **do**

20:              **for** $y \in \mathrm{Dom}(Y)$ **do**

21:                  Perform $a = \mathrm{do}(Y = y)$ for $B$ times, collect $R_1^a, \ldots, R_B^a$.

22:                  **if** $|\hat{R} - \frac{1}{B}\sum_{b=1}^B R_b^a| > \Delta/2$ **then**

23:                      RewardBranch $\leftarrow B_{\widetilde{\mathcal{T}}_0}^{v_c(t):Y}$.                 by Assumption 3.

24:                  **end if**

25:              **end for**

26:          **end for**

27:          **if** RewardBranch is None. **then**

28:              **return** $v_c(t)$. //$v_c(t)$ is an ancestor of $X_R$ but none of its children is, thus, $v_c(t)$ itself is $X_R$.

29:          **end if**

30:          $q_{t+1}(X) \leftarrow 0$ for $X \notin$ RewardBranch and $q_{t+1}(X) \leftarrow 1$ for $X \in$ RewardBranch.   //only the variables in RewardBranch might be $X_R$.

31:      **end if**

32:      normalize $q_{t+1}(\cdot)$, $t \leftarrow t + 1$.

33: **end while**

---

---

**Algorithm 5** CN-UCB for Causal Forest

---

1: **Input:** essential graph $\mathcal{E}(D)$, $K, \varepsilon, \Delta, B, T_2$, observational probabilities $P(\mathcal{X})$.

2: Perform do() for $B$ times, collect $R_1, \ldots, R_B$, $\hat{R} \leftarrow \frac{1}{B}\sum_{b=1}^B R_b$.

3: **for** $\mathcal{T}_0$ in $\mathrm{CC}(\mathcal{E}(D))$ **do**

4:      **Stage 1:** Find a directed subtree that contains $X_R$: $\widetilde{\mathcal{T}}_0 \leftarrow$ Find Sub-tree$(\mathcal{T}_0, K, B, \varepsilon, \Delta, \hat{R})$.                                              //call Algorithm 3.

5:      **if** $\widetilde{\mathcal{T}}_0$ is not empty **then**

6:          **Stage 2:** Find the key node that generates the reward: $X_R \leftarrow$ Find Key Node$(\widetilde{\mathcal{T}}_0, K, B, \Delta, \hat{R})$.                     //call Algorithm 4.

7:          **Stage 3:** Apply UCB algorithm on $\mathcal{A}_R = \{\mathrm{do}(X_R = k) \mid k = 1, \ldots, K\}$ for $T_2$ rounds.

8:      **end if**

9: **end for**

---

---

**Algorithm 6** Find Central Node

---

**Input:** Undirected tree $\mathcal{T}$ with some distribution $q$ over the nodes $i = 1, \ldots, n$.

1: Choose a node $v$ from $X_1, \ldots, X_n$. Find neighbors $N_{\mathcal{T}}(v)$.
2: **while** $\max_{j \in N_{\mathcal{T}}} q(B_{\mathcal{T}}^{v_c : X_j}) \geq 1/2$ **do**
3: $\quad v \leftarrow \mathrm{argmax}_{j \in N_{\mathcal{T}}} q(B_{\mathcal{T}}^{v_c : X_j})$
4: **end while**

**Output:** Central node $v_c = v$.

---

---

**Algorithm 7** Find Sub-junction-tree

---

1: **Input:** graph $G$, junction tree $\mathcal{T}_G, K, B, \varepsilon, \Delta, \hat{R}$.
2: **Initialize:** $t \leftarrow 0, q_0(C) \leftarrow 1/|\mathcal{T}_G|$ for all $C \in \mathcal{T}_G$, found $\leftarrow$ False (indicating whether an ancestor of $X_R$ is found or not).
3: **while** found is False **do**
4: $\quad$ Identify central node $C_c(t) \leftarrow$ Find Central Node$(\mathcal{T}_G, q_t(\cdot))$. $\qquad$ //call Algorithm 6.
5: $\quad$ Initialize the edge direction between $C_Y$ and $C_c(t)$ as $C_Y \to C_c(t)$ by:
$\quad$ direction$(C_Y) \leftarrow$ up, for all $C_Y \in N_{\mathcal{T}_G}(C_c(t))$.
6: $\quad$ Clique Intervention $(G, C_c(t), B)$ with collected interventional samples: $R_b^{\mathrm{do}(Z=z)}, Y_b^{\mathrm{do}(Z=z)}$
$\quad$ for $Z \in V(C_c(t)), z \in [K], Y \in V(G), b = 1, \ldots, B$. $\qquad$ //call Algorithm 9.
7: $\quad$ $\hat{P}(Y = y|\mathrm{do}(Z = z)) \leftarrow \frac{1}{B} \sum_{b=1}^{B} \mathbb{1}_{\{Y_b^{\mathrm{do}(Z=z)}=y\}}$ for $Z \in C_c(t), z \in [K], Y \in V(G) \setminus \{Z\}, y \in [K]$.
8: $\quad$ **if** $|\hat{R} - \frac{1}{B}\sum_{b=1}^{B} R_b^{\mathrm{do}(Z=z)}| > \Delta/2$ for some $Z \in C_c(t), z \in [K]$ **then**
9: $\quad\quad$ found $\leftarrow$ True. $\qquad$ //by Assumption 3.
10: $\quad$ **end if**
11: $\quad$ **for** $C_Y \in V(N_{\mathcal{T}_G}(C_c(t)))$ **do**
12: $\quad\quad$ **if** $\forall Z \in C_Y \cap C_c(t), \forall Y \in C_Y \setminus C_c(t), \exists z, y$, s.t. $|P(Y = y) - \hat{P}(Y = y \mid \mathrm{do}(Z = z))| > \varepsilon/2$ **then**
13: $\quad\quad\quad$ direction$(C_Y) \leftarrow$ down. $\qquad$ //by Assumption 2 and Definition 8, the edge direction is updated as $C_c(t) \to C_Y$.
14: $\quad\quad$ **end if**
15: $\quad$ **end for**
16: $\quad$ **if** found is True **then**
17: $\quad\quad$ **return** sub-junction-tree $\widetilde{\mathcal{T}}_G$ induced by $\{C \mid C \notin B_{\mathcal{T}_G}^{C_c(t):C_Y}$, where direction$(C_Y) =$ up$\}$
18: $\quad$ **end if**
19: $\quad$ **for** $C_Y \in N_{\mathcal{T}_G}(C_c(t))$ **do**
20: $\quad\quad$ **if** direction$(C_Y) =$ down **then**
21: $\quad\quad\quad$ $q_{t+1}(X) \leftarrow 0$ for $X \in B_{\mathcal{T}_G}^{C_c(t):C_Y} \cup \{C_c(t)\}$. $\qquad$ //cliques that cannot contain $X_R$.
22: $\quad\quad$ **else**
23: $\quad\quad\quad$ $q_{t+1}(X) \leftarrow 1$ for $X \in B_{\mathcal{T}_G}^{C_c(t):C_Y}$. $\qquad$ //cliques that may contain $X_R$.
24: $\quad\quad$ **end if**
25: $\quad$ **end for**
26: $\quad$ normalize $q_{t+1}(\cdot), t \leftarrow t + 1$.
27: **end while**

---

**Algorithm 8** Find Key Clique

---

1: **Input:** graph $G$, directed sub-junction tree $\widetilde{\mathcal{T}}_G$, $K, B, \Delta, \hat{R}$.
2: **Initialize:** $t \leftarrow 0$, $q_0(C) \leftarrow 1/|\widetilde{\mathcal{T}}_G|$ for clique $C$ in $\widetilde{\mathcal{T}}_G$.
3: **while** $q_t(C) > 0$ for more than one clique $C$ in $\mathcal{T}_{G_0}$ **do**
4:     Identify central clique $C_c(t) \leftarrow$ Find Central Node($\widetilde{\mathcal{T}}_G, q_t(\cdot)$).         //call Algorithm 6.
5:     Initialize direction $\leftarrow$ up. //Indicating whether the true $X_R$ is towards the upstream direction of $C_c(t)$ (not include $C_c(t)$) or downstream direction of $C_c(t)$ (include $C_c(t)$).
6:     Clique Intervention($G, C_c(t), B$) and collect reward data $R_1^{\mathrm{do}(Z=z)}, \ldots, R_B^{\mathrm{do}(Z=z)}$, for $Z \in C_c(t), z \in [K]$.         //call Algorithm 9.
7:     **if** $|\hat{R} - \frac{1}{B}\sum_{b=1}^{B} R_b^{\mathrm{do}(Z=z)}| > \Delta/2$ for some $Z \in C_c(t), z \in [K]$ **then**
8:         directon $\leftarrow$ down.         //by Assumption 3.
9:     **end if**
10:     **if** direction = up **then**
11:         $q_{t+1}(C) \leftarrow 0$, for clique $C \in B_{\widetilde{\mathcal{T}}_G}^{C_c(t):C_Y}$ if $C_Y$ satisfies: $C_c(t) \to C_Y$ in $\widetilde{\mathcal{T}}_G$.
12:         $q_{t+1}(C) \leftarrow 1$, for the remaining cliques.
13:     **else**
14:         RewardBranch $\leftarrow$ None.
15:         **for** $C_Y \in \mathrm{Ch}_{\widetilde{\mathcal{T}}_G}(C_c(t))$ **do**
16:             Clique Intervention($G, C_Y, B$) and collect reward data $R_1^{\mathrm{do}(Y=y)}, \ldots, R_B^{\mathrm{do}(Y=y)}$, for $Y \in V(C_Y), y \in [K]$.         //call Algorithm 9.
17:             **if** $|\hat{R} - \frac{1}{B}\sum_{b=1}^{B} R_b^{\mathrm{do}(Y=y)}| > \Delta/2$ for some $Y \in V(C_Y), y \in [K]$ **then**
18:                 RewardBranch $\leftarrow B_{\widetilde{\mathcal{T}}_G}^{C_c(t):C_Y}$.
19:                 **Break**
20:             **end if**
21:         **end for**
22:         **if** RewardBranch is None **then**
23:             **return** $C_c(t)$. //One of the variables in $C_c(t)$ is an ancestor of $X_R$, but none of $C_c(t)$'s children contains an ancestor of $X_R$. Thus, $C_c(t)$ contains $X_R$.
24:         **else**
25:             $q_{t+1}(C) \leftarrow 1$, for $C \in$ RewardBranch.     //only cliques in RewardBranch may contain $X_R$.
26:             $q_{t+1}(C) \leftarrow 0$, for $C \notin$ RewardBranch.     //cliques not in RewardBranch do not contain $X_R$.
27:         **end if**
28:     **end if**
29:     normalize $q_{t+1}(\cdot)$, $t \leftarrow t + 1$.
30: **end while**

---

**Algorithm 9** Clique Intervention

---

1: **Input:** Graph $G$, Clique $C$, $B$.
2: **for** $Z \in V(C)$ **do**
3:     **for** $k = 1, \ldots, K$ **do**
4:         **for** $b = 1, \ldots, B$ **do**
5:             Perform intervention $\mathrm{do}(Z = k)$.
6:             Collect interventional data for reward and other variables on the graph: $R_b^{\mathrm{do}(Z=k)}$, $Y_b^{\mathrm{do}(Z=k)}$ for $Y \in V(G)$.
7:         **end for**
8:     **end for**
9: **end for**