# OpenReview forum: "Causal Bandits with Unknown Graph Structure"
_NeurIPS.cc/2021/Conference — NeurIPS 2021 Poster_

### Official Review · Reviewer_25sC · 2021-07-16

**Rating:** 7
**Confidence:** 4

**Summary:**

This paper gives low-regret algorithms for the causal bandits problem when the graph structure is unknown. The main contribution is in combining use of some recent results on efficient causal structure discovery with traditional UCB approaches. Some minimax results are given to demonstrate the necessity of the assumptions made.

**Limitations And Societal Impact:**

No broader impact statement is given here but it isn't really necessary in this case.  The scope of the work was generally clear and some nice directions for future work were given.

**Main Review:**

### Originality
+ The use of the causal discovery results by Greenewald et al. (19) and Squires et al. (20) within the context of a targeted structure discovery problem is a very nice idea. An assumption on there being some kind of tree structure (either a graph that is a forest or junction tree), allows the proposed algorithm to be more efficient than a naive application of these structure learning approaches, because they disregard parts of the tree as soon as they are found not relevant to the reward.
+ To my knowledge this is the only work on causal bandits where the causal graph structure is unknown.

### Quality
+ The assumptions are intuitive and the results of section 6 reinforce their importance.
- No consideration is given regarding the computational complexity of selecting each intervention. This seems like something that could at least be commented on for the various stages of Algorithm 1.
- Would be nice to see a comment on how well these ideas might extend to continuous random variables, in particular linear bandits.
- Other works I've seen on this area have included some experimental results on simulated data. I think there would certainly be some value in those here: demonstrating practical implementation of the proposed algorithms; considering the effect of graph structure on performance; performance when the identifiability assumptions break; performance when the intersection-incomparable assumption doesn't hold on the clique graph. Relevant baselines would be UCB and existing algorithms that get given the graph structure.

### Clarity
+ Description of the results, especially for tree graphs, is very clear and easy to follow. Figure 1 is great for following the algorithm explanation.
- It could be made more clear exactly what assumptions are allowing us to improve over UCB in terms of sample efficiency. It seems that it is because of both the assumptions on having some non-degenerate graph structure and that we get side feedback on the values of nodes, which is not present in the typical bandit problem. These assumptions are just as important to understanding the results as the well emphasised assumptions on identifiability.
- I think it is worth acknowledging that whilst previous works are limited in assuming knowledge of the causal graph, they do also address various settings which would be considered natural extensions to this paper. For example Lu et al. (19) consider interventions on multiple variables simultaneously, and the linear bandit setting.
- line 350: does -> do.

### Significance
+ Causal bandits is a problem area that has received a fair bit of attention in the last few years, and good guarantees for the setting of an unknown graph structure is a significant contribution to the field.
+ The minimax results also offer progress on understanding what assumptions are necessary to achieve interesting regret bounds for causal bandits when the graph structure is not known.

**Time Spent Reviewing:**

6

---

> ### Author Response · Authors · 2021-08-10
> **Response to Reviewer 25sC**
>
> Thanks for your constructive comments. We are very glad you thought this a significant contribution to the field. Please see below for our responses.
>
> 1. Computational complexity:
>
> We provided sample complexity of interventions for stage 1 and 2 in Algorithm 1 (Lemma 1 in Appendix A.1). In stage 1 and 2, all computations for each intervention are for calculating interventional probabilities and updating the probabilities $q_t()$ that define the central node (see details in Algorithm 3 and 4). The computational complexity for Algorithm 1 in stage 1 and 2 is at most $\tilde{O}(K^2\max(1/\Delta^4, 1/\epsilon^4)d+n)$, ignoring poly-log terms, and stage 3 is just UCB. Causal forest and general graph settings are similar. We will add discussion on this.
>
> 2. “Generalization to continuous random variables”:
>
> Our methods can be generalized to continuous variables under suitable identifiability assumptions like Assumption 2 and 3. Since the whole idea of our paper is to cut branches using the reward signal and the causal relations among variables, we believe as long as the branch cutting step can be performed, similar results should hold. This is an interesting next step to study.
>
> 3. “Experiments and discussions on breaking several assumptions”:
>
> Our work mainly focuses on developing methodologies for the unknown graph causal bandit setting. Since this is the first work on this topic, we didn’t discuss practical issues such as breaking several assumptions and conducting comparisons in experiments. What you suggested are indeed worth pursuing in future works.
>
> 4. "It could be made more clear exactly what assumptions are allowing us to improve over UCB in terms of sample efficiency...":
>
> You are right. The graph structure and the side feedback on the values of nodes together with the identifiability assumptions enable us to do better than a UCB algorithm. We will explain these clearly. Thanks for your suggestion.
>
> 5. “Acknowledging previous works on multiple variable interventions and the linear bandit setting…”:
>
> We agree that aforementioned settings in existing works are interesting extensions. We will definitely look into these in our next steps.
>
> 6. We will fix the typos carefully, thanks for pointing out!

---

> > ### Comment · Reviewer_25sC · 2021-08-15
> > **25sC Review Followup**
> >
> > Thank you for your response. A paper by de Kroon et al. [1] takes a different approach, based upon separating sets, to tackle the causal bandit problem with unknown graph structure. Please could you comment on the relation of your work to this paper? I believe it should be considered relevant literature since it is the only other paper attempting causal bandits with unknown graph structure.
> >
> > [1] de Kroon, Arnoud AWM, Danielle Belgrave, and Joris M. Mooij. "Causal discovery for causal bandits utilizing separating sets." arXiv preprint arXiv:2009.07916 (2020).

---

> > > ### Author Response · Authors · 2021-08-16
> > > **Response to 25sC Review Followup**
> > >
> > > Thanks for pointing out de Kroon et al. (2020). Our work differs from theirs in the following aspects:
> > >
> > > 1. First of all, the assumptions and approaches are different: In de Kroon et al. (2020), they assume there exists a set of non-intervenable variables, denoted as S, that d-separate the interventions and the reward. Their algorithm takes an existing causal discovery method to find S, based on which they derive a UCB (or Thompson Sampling)-type algorithm. On the other hand, our work assumes all variables can be intervened (except for the reward), and we utilize the reward signal to learn the direct causes of the reward directly and then apply the UCB algorithm on those direct causes.
> > >
> > > 2. Secondly, our work focuses on the theoretical aspect. We obtain improved regret guarantees over the normal UCB. de Kroon et al. (2020) focuses more on the empirical aspect. Theoretically, de Kroon et al. (2020) only prove that their algorithm has the same regret bound as normal UCB (see their Theorem 4.1).
> > >
> > > We will add discussions about this paper.

---

> > > > ### Comment · Reviewer_25sC · 2021-08-26
> > > > **Follow-up questions**
> > > >
> > > > Thank you for your response. There are some concerns in the reviews in terms of significance. I'll try to pose some questions to allow you to address them
> > > >
> > > > Do you think the structural assumptions on the DAG made in this paper are general enough to motivate a practical algorithm? If so, what would this setting be? It seems like if the true DAG doesn't fit the structural assumption, the method proposed could do arbitrarily badly. Therefore, it seems you would need to select a setting where causal bandits are relevant, and you can confidently assume a tree-like structure for the true DAG.
> > > >
> > > > It seems like the most natural candidate would be bipartite graphs occurring in biology. The paper mentions this and I took a look at the citations given on line 132. Kontou et al. (2016) looks at gene-disease networks. These are bipartite, but the directions are all obvious (gene -> disease), which makes the method we're reviewing irrelevant. I looked briefly at Pavlopoulos et al. (2018) too, and in all the examples there I see the same thing.
> > > >
> > > > If it is not the case that the proposed method could inspire a practical approach without further work, do you think the proposed approach provides some basis for tackling the causal bandit problem with a more general class of unknown DAGs? The proposed method heavily relies upon an approach that makes use of a tree-like graph structure through central-node arguments. Therefore it seems like the significance of the paper is heavily reliant on how practical the setting where we must assume a tree-like structure is.
> > > >
> > > > Please let me know your thoughts on these points. Thank you.

---

> > > > > ### Author Response · Authors · 2021-08-27
> > > > > **Response to 25sC Follow-up questions**
> > > > >
> > > > > Thanks for raising these questions. We address your questions below, and we will also add more discussions in the final version.
> > > > >
> > > > > 1: Our algorithm in Section 5 works for general DAGs.
> > > > >
> > > > > We want to clarify that our algorithm for general DAGs in Section 5 does not require tree-structure assumptions. We start with causal trees and causal forests to illustrate the central node intervention idea, but this idea is not limited to tree-like DAGs.
> > > > >
> > > > > Specifically, in Algorithm 2, we search for the reward generating variable ($X_R$) in every chain component of the essential graph.
> > > > > For each component, we construct a junction tree, and then search for a directed sub-junction-tree that contains $X_R$ (line 5 of Algorithm 2).
> > > > > For the directed sub-junction-tree $\tilde{\mathcal{T}}_G$ (of a certain component) that contains $X_R$, we search for the clique in $\tilde{\mathcal{T}}_G$ that contains $X_R$ (line 7 of Algorithm 2).
> > > > > In Algorithm 2, the idea of central node interventions is generalized to central clique interventions.
> > > > >
> > > > >  We use the fact that every chain component of the essential graph is a chordal graph, and one can always construct a junction tree over a chordal graph (see [1]).
> > > > > Such graph-theoretic properties guarantee that Algorithm 2 can be applied to general DAGs (doesn’t have to be tree-like).
> > > > >
> > > > > [1] Squires et al. (2020) Active Structure Learning of Causal DAGs via Directed Clique Trees (NeurIPS 2020)
> > > > >
> > > > > 2. Real-world Causal Forest Example:
> > > > >
> > > > > For example, in gene-protein networks (which are bipartite graphs), genes can cause protein expressions, and the expressed proteins can block or activate other genes as well [2]. These proteins are called gene activator proteins [3].  In this setting, for each gene-protein pair, we need to learn its direction.
> > > > >
> > > > > [2] Greenewald et al. (2019) Sample Efficient Active Learning of Causal Trees (NeurIPS 2019)
> > > > >
> > > > > [3] https://en.wikipedia.org/wiki/Activator_(genetics)

---

> > > > > > ### Comment · Reviewer_25sC · 2021-08-27
> > > > > > **response and question clarification**
> > > > > >
> > > > > > Thank you for your answers. I have the following responses:
> > > > > >
> > > > > > 1) Sorry for my lack of specificity. If you could address my original question but replace "has a tree-like structure" with "satisfies the intersection-incomparable property" that would be great. My understanding is that this algorithm does not hold for general graph structures but requires graphs to satisfy the intersection-incomparable property.
> > > > > >
> > > > > > 2) The citation you give is another structure learning paper, which in turn cites Kontou et al. (2016) on this matter. In that paper I only see discussion of gene-disease networks and not gene-protein networks.

---

> > > > > > > ### Author Response · Authors · 2021-08-28
> > > > > > > **Reponse to Reviewer 25sC on the clarified questions**
> > > > > > >
> > > > > > > Thanks for your clarifications. We are very happy to discuss.
> > > > > > >
> > > > > > > 1. Intersection-incomparable assumption in Section 5:
> > > > > > >
> > > > > > > In order to obtain an improved regret bound in Theorem 3, we do need the intersection-incomparable assumption. This assumption guarantees there's no v-structure over cliques in any junction tree, so that our algorithm can eliminate at least half of the cliques by interventions over the central clique. (See detailed discussion in line 317 - 327)
> > > > > > >
> > > > > > > Without this assumption, Algorithm 2 may stop at line 7 without finding the clique that contains the reward generating variable $X_R$.
> > > > > > > For example, suppose A->B<-C<-D is a directed junction tree with v-structure where A/B/C/D are cliques and $X_R$ is contained in A. Suppose we intervene on C and don’t observe any reward change.
> > > > > > > Then according to Algorithm 2, once we figure out B<-C and C<-D, the algorithm will cut off A/B/C from future searching and will never find a clique that contains $X_R$.
> > > > > > > In this case, Algorithm 2 will not output an incorrect clique but will stop at line 7, and the learner can use any standard bandit algorithm by ignoring causal information.
> > > > > > >
> > > > > > > Note that one cannot always do better than standard bandit algorithms by using causal information. See line 42 - 46. Some structural assumptions are needed to guarantee improved regret bounds.
> > > > > > >
> > > > > > > 2. About gene-protein bipartite graph:
> > > > > > >
> > > > > > > Using bipartite graph to model a gene-protein (sometimes called RNA-protein / DNA-protein) relationship is common. For example, [1] and [2] discussed the RNA-protein bipartite network.
> > > > > > >
> > > > > > > [1] Ge et al. (2016) A Bipartite Network-based Method for Prediction of Long Non-coding RNA–protein Interactions
> > > > > > > [2] Zhao et al. (2018) The Bipartite Network Projection-Recommended Algorithm for Predicting Long Non-coding RNA-Protein Interactions

---

### Official Review · Reviewer_Em8x · 2021-07-16

**Rating:** 8
**Confidence:** 5

**Summary:**

Authors propose using the existing adaptive causal discovery algorithms to identify the reward variable in a causal graph efficiently and then run the existing bandit algorithms on this very small space of actions (interventions). They use the ideas from central node algorithm of Kristjan et al.: Identify a central node, either eliminate ~half the graph or find the subtree including the target node. They then have to check each child separately since either one of them can be the target node - which also serves as a method to check if central node itself is the target node. This takes ~dlogn interventions which can easily be converted to sample-complexity with the resorted assumptions.


**Limitations And Societal Impact:**

Yes

**Main Review:**

The two assumptions that are somewhat restrictive are the tree and in the next section intersection comparability assumption which was made by Squires et al. and that there is only a single node that affects the reward function. Despite these, I think this is a very nicely done first effort in this space of combining causal discovery with bandit algorithms.

Authors also show necessity of the assumptions by constructing counterexamples based on Lattimore's results. I think these are very important and insightful results and I am glad the authors included those.

Minor comments:
- Footnote seems to be hinting at that single X_R assumption is wlog. Is this true? Please elaborate on what fails or if it is. Discussion at the end seems to suggest this is only a tentative idea. Please clarify in the text as well.
- Causal faithfulness definition is not precise, please check. It should say "... has ONLY the independence relations". missing only completely changes meaning here.
- Directed tree assumption is indeed motivated by essential graph here but this is not explicitly stated in page 3 which makes the assumption look more restrictive than it actually is.
- Essential graph definition in page 3 has an issue: It doesn't only involve edges that are part of v-structures but also those implied by Meek rules. E.g. X->Y->T + Z->T is the DAG and the essential graph since Z->T can be identified by the first Meek rule.
- abstract: MAB bandit->MAB.


**Time Spent Reviewing:**

3

---

> ### Author Response · Authors · 2021-08-10
> **Response to Reviewer Em8x**
>
> Thanks for your constructive comments! Please see below for our response
>
> 1. Multiple reward-generating variables:
>
> We have the following idea for the case where the reward has multiple direct causes (the ‘former’ one in your review).
> We will have to generalize Assumption 3 to hold on interventions on any ancestor of each of the direct causes. That is to say for any variable $X$ that is an ancestor of certain direct cause of the reward, we have $|E[R|do(X=x)] - E[R|do()]| > \Delta$ for some $x$ in the domain of $X$. Then our approach can be generalized to this setting by running multiple times.
> Specifically, in Algorithm 1, we don’t stop after stage 2 finds a reward generating variable, say $X_{R1}$, because it is only one of the direct causes.
> By construction of stage 2 (Algorithm 4), we know except for $X_{R1}$ itself, none of its descendants is a direct cause of the reward (we always check children first). We can fix the values of all variables in the subtree $\mathcal{T_1}$ induced by $X_{R1}$ (as the root).
> Then we re-run Algorithm 1 on the remaining graph (still a tree): original tree cut by subtree $\mathcal{T_1}$, and find a second direct cause of the reward. This procedure can be repeated until no further reward-generating variable can be found.
> Above idea is a way to find multiple reward generating variables, but it’s also possible to come up with more efficient ways. For example, instead of finding the direct causes one by one, it will be interesting to develop methods that can find several of them simultaneously. We think this setting itself is also interesting and worth studying as an independent work.
>
> 2. We will rephrase to motivate the causal tree setting more clearly, thanks for your suggestion.
>
> 3. We will correct the essential graph definition about v-structures and the typos in stating causal faithfulness and the abstract. Thanks for pointing these out. We note that these do not affect our results.

---

> > ### Comment · Reviewer_Em8x · 2021-08-23
> > **rebuttal response**
> >
> > Hi, thank you for your response! I recommend that the authors include the explanation on how to handle the multiple reward-generating variables case in the paper or in the appendix in the camera-ready version.

---

> > > ### Author Response · Authors · 2021-08-23
> > > **Thanks for your suggestion!**
> > >
> > > Thanks for your suggestion, we will add the explanations!

---

### Official Review · Reviewer_sHAH · 2021-07-17

**Rating:** 3
**Confidence:** 4

**Summary:**

The authors present a novel approach to tackle so-called causal bandit problems where the structure of the underlying graph is unknown. The algorithm is based on a ‘central node’ approach to find the direct parent of the reward as soon as possible. Under certain restrictions various regret bounds are derived for this algorithm that show the asymptotic behaviour outperforms standard multi-arm bandits by a significant margin, though no experimental verification is provided.


**Ethical Concerns:**

No ethical concerns.

**Limitations And Societal Impact:**

No direct societal impact.

**Main Review:**

The problem considered in the paper is indeed challenging and relevant, and has so far not received the attention it deserves.

Unfortunately, despite some nice results on regret bounds, the execution in the paper falls far short of what is promised at the outset. The end result is very disappointing and I consider it a missed opportunity.

On the plus side, the paper technically does provide a causal bandit approach for unknown causal structure that outperforms standard MABs, and it does offer some nice results on regret bounds. However it does so by invoking such strong restrictions and assumptions that it effectively kills the problem it set out to solve.

- Essentially it is a standard ‘balanced split’ search method that is then presented as a ‘causal bandit approach’ by restricting the class of allowable causal models so far that the proposed search method works. These restrictions/assumptions reduce the problem to a trivial case (tree like where all ancestors have measurable impact on the reward) with little hope for any real-world situation to fit the problem. And there is little hope of adapting the method to more realistic causal structures, as the core of the method has little to do with causality or causal structures in the first place.

- The ‘with unknown graph structure’ from the title turns out to involve full knowledge of the equivalence class representation of the model after all. And then it is supposed to be of a very specific type as well.

- The ‘causal sufficiency’ part of assumption 1 is disappointing as in practice the larger the graph the more likely we are missing some relevant confounders. For a principled causal bandit approach this should not erven be necessary, but ok. Causal faithfulness is not needed for the current task (provided assumption 2 holds).

- The assumed ‘causal tree / forest’ structures are extremely limiting cases of the general class of all  possible causal structures (with or without sufficiency). Worse, the then introduced ‘general class of causal graphs’  (section 5) turns out to be almost as restrictive. In other words: only if the causal bandit system under investigation happens to satisfy an extreme configuration can we apply the method presented here. And for large graphs the chance that any random causal structure will satisfy this condition is virtually negligible.

- Assumption 3 however is the most extreme and completely unrealistic in practice, especially if the (assumed) tree graph is large. It states that all ancestors of the reward function have a measurable impact on it. However, in practice long directed chains tend to result in extremely weak dependencies that will remain undetectable even with huge amounts of data.

- Even worse: under the extremely strong assumptions in the paper the problem can be solved much more quickly than with the algorithm presented, because as soon as there is an intervention on ANY of the ancestors of the reward function in stage 2 of the algorithm, its single direct parent can be found immediately by virtue of assumption 3: it will simply be the variable that shows the strongest dependence with the actual reward. No other tests are needed … which also indicates how niche and limiting the current setup is.


originality: novel approach but still fairly derivative
quality: main results (Theorems 1-3 and CN-UCB algorithm) are technically correct. However the method is seriously miss-sold as a ‘general approach under mild conditions’, and despite the ‘unknown graph structure’ claim in the title, the first line of the algorithm starts with having the graphical equivalence class representation of the true causal structure: that is just not good enough. Also no experimental evaluation is provided.
clarity: writing is ok, with some minor flaws. Some good examples, but far too much emphasis on the straightforward ‘tree/forest’ case (7 pages) leaving nowhee near enough space for the general case (5.2) and lower bounds (section 6).
significance: the approach is too niche and weirdly restrictive, making it unlikely to generate much interest from the (causal) bandit community. In fact it hardly deserves the moniker ‘causal’ at all, and is in many respects a missed opportunity.


minor comments:
p3.125: ‘Every directed edge in E(D) is within a v-structure’ => this is wrong. It is true that every v-structure corresponds to a directed edge in E(D), but other invariant directed edges are also possible.

**Time Spent Reviewing:**

4 hrs

---

> ### Author Response · Authors · 2021-08-10
> **Response to Reviewer sHAH**
>
> We thank the reviewer for the detailed comments. We address each concern below.
>
> 1. “Essentially…...” and “The assumed ‘causal tree / forest’ structures are extremely limiting…....” :
>
> We disagree that our method is extremely limiting because we start with causal trees or causal forests. In causal discovery, it is standard to start with tree structures, building on which can be extended to forests or more general graphs using junction tree ideas (Squires et al. (2020)). We follow the same order in designing our methods. Even causal trees and causal forests have wide applications in biology and epidemiology (Greenewald et al., 2019; Burgos et al., 2008; Kontou et al., 2016; Pavlopoulos et al., 2018).
> We would like to also mention that existing causal bandit works with regret analysis all require full knowledge of the directed causal graph. Some of them even require the knowledge of certain probabilities along the graph (See e.g. Lattimore et al., 2016; Lu et al., 2019; Nair et al., 2021). Our paper is the first one that relaxes this kind of strong assumption and should be considered as a realistic step forward for practical problems.
>
> 2. “Known essential graph is not good enough”:
>
> Our work focuses on intervention design in the bandit setup, and like most of the intervention design works on causal discovery, we take the essential graph as the input of our algorithms. We would like to especially mention that this is not a strong assumption, because the essential graph can be estimated accurately from enough observational data using well-developed methods. Observational data is much cheaper than interventional data and is usually not a concern and using the essential graph as input is quite standard in interventional design studies. See references below. We will mention this in the abstract, thanks for your suggestion.
>
> [1] He, Y.-B. and Geng, Z. (2008)
> Active learning of causal networks with intervention experiments and optimal designs.
> Journal of Machine Learning Research, 9(Nov):2523–2547.
>
> [2] Squires, C., Magliacane, S., Greenewald, K., Katz, D., Kocaoglu, M., and Shanmugam, K. (2020)
> Active structure learning of causal dags via directed clique tree.
> 34th Conference on Neural Information Processing Systems (NeurIPS 2020)
>
> [3] Greenewald, K., Katz, D., Shanmugam, K., Magliacane, S., Kocaoglu, M., Adsera, E. B., and Bresler, G. (2019)
> Sample efficient active learning of causal trees.
> 33rd Conference on Neural Information Processing Systems (NeurIPS 2019)
>
> 3. “Causal Sufficiency is disappointing and a principled causal bandit approach does not need this.”:
>
> Causal sufficiency is a standard assumption in causal discovery studies, e.g. Squires et al. (2020) and Greenwald et at. (2019). It is also necessary in our setting since otherwise the directions of edges will be hard to identify. Even though this assumption is not needed in some other causal bandit approaches, but it is because they have full knowledge of the directed causal graph and they don’t need to learn the graph edge directions. Compared to existing works' known full graph assumption, we believe causal sufficiency is mild.
>
> 4. Assumption 3 unrealistic:
> In our lower bound result (theorem 5), we prove that without Assumption 3, even if assumption 1 and 2 hold, one can still not do better than standard multi-armed bandit algorithms. Thus, Assumption 3 or other similar assumption is needed for developing methods that outperform standard multi-armed bandit approaches.
>
> 5. “Even worse: under the extremely strong assumptions in the paper the problem can be solved much more quickly…”:
>
> It is not correct that an intervention on any of the ancestors of the reward can deduce the true reward-generating variable by Assumption 3. Here is a counter-example: the true structure is $X_1\rightarrow X_2$ and $X_2$ the direct cause of the reward. Suppose we intervene on $X_1$ (an ancestor of reward), by Assumption 3 we can only know the reward-generating variable is one of the descendants of $X_1$, which includes $X_1$ as well. This is because Assumption 3 only guarantees the reward behaves differently under empty intervention and interventions on $X_1$. We can’t determine whether the reward is generated by $X_1$ or $X_2$.
>
> We agree that the essential graph does not only contain directed edges in a v-structure. Thanks for your correction! We will correct this carefully. We note that this does not affect our results.

---

> ### Author Response · Authors · 2021-08-23
> **HAS OUR RESPONSE ADDRESSED YOUR CONCERNS?**
>
> Hi Reviewer sHAH, we would be grateful if you can confirm whether our response has addressed your concerns, and let us know if any issues remain. To recap our response, we explained:
> 1. why our methods are not limiting (even causal trees and causal forests have wide applications in biology and epidemiology);
> 2. knowing the essential graph is not a strong assumption and many other works also assume this;
> 3. causal sufficiency is a standard assumption in causal discovery studies. Some existing causal bandit works do not require this because they need the full knowledge of the causal graph;
> 4. why Assumption 3 is necessary (proved in the lower bound section);
> 5. it is not correct that an intervention on any of the ancestors of the reward can deduce the true reward-generating variable by Assumption 3.

---

### Official Review · Reviewer_d51G · 2021-07-28

**Rating:** 7
**Confidence:** 4

**Summary:**

The paper presents a causal bandit algorithm for graphs with unkown causal structure (but no unobserved confounders). It is based on a couple of recent results in learning causal graphs using interventions, and previous causal graph work.

**Limitations And Societal Impact:**

Societal impact: similar to other papers in the field, so I don't think much new has to be said.


**Main Review:**

Authors provide an algorithm for causal bandits with logarithmic regret with a graph of unknown structure. Unlike causal graph setting, where the goal is to learn the whole graph, in causal bandit setting it is unnecessary, which is one major difference between this and previous work on causal graph structure. I am not aware of the previous causal bandit work that can achieve this on unknown graphs. Thus combining the two approaches is the novelty of the paper.

 Their assumptions are:

-Standard graphical causality assumptions

-No hidden confounders (also standard -- while most applications will have hidden confounders, proving anything in their presence gets much harder).

-Underlying graph's clique tree has intersection-incomparible property. This last assumption is the least common of the three, and something that would be great to relax if possible or further justify. It is also present in previous work on causal graphs which this work is based on.

I did not check the proofs in the Appendix carefully, and assumed for this review that proofs hold. They seem to hold "intuitively", so I would be surprised if they do not, but can not vouch for that.


**Time Spent Reviewing:**

10

---

> ### Author Response · Authors · 2021-08-10
> **Response to Reviewer d51G**
>
> We thank the reviewer for the positive feedback.
>
> For the general causal graph setting, yes, we follow the intersection-incomparable assumption in the causal discovery work (Squires et al. (2020)). We agree that further relaxing or justify it is an interesting next step. Thanks for your suggestion.

---

### Official Review · Reviewer_xwY7 · 2021-08-01

**Rating:** 7
**Confidence:** 3

**Summary:**

This paper investigates the problem of 'causal bandits,’ a sequential decision problem where the agent can intervene on a causal DAG (setting the value of a variable) and receives a reward associated with an unknown variable’s value.  Unlike prior works, in this paper, only partial prior knowledge of the structure is known (essential graph).  In addition to the reward, the agent also observes the realization of all the variables; that information with the essential graph helps inform the agent of where in the graph the reward-generating variable is.   Although the agent need not learn the full causal structure for the task, prior work on causal structure learning provides an algorithmic foundation for approaching the causal bandits problem.  Regret bounds for the algorithms are provided for trees, forests, and then general graphs (which satisfy certain topological constraints).  Lower bounds are also provided.

**Limitations And Societal Impact:**

Yes (while there are potential applications of this idea to the sciences, this work itself is theoretical)

**Main Review:**

Major Comments
•	Overall I found the paper quite interesting.  As the authors are upfront about, the algorithm design largely builds on central-node procedures for learning the causal structure.  Especially for trees and forests, that approach is at a high-level intuitive and the authors demonstrate effective.  As the authors point out, for the causal bandit problem, you only need to learn enough about the structure to identify with high confidence what region of the graph contains that variable; eg it is suboptimal to have one stage where you fully learn the graph and then search for reward-generating variable.
•	In terms of contribution, if the focus were just on trees, that might not have been enough, but the authors extend the algorithms to a class of general graphs, though point out limitations of central-node procedures in that case.
•	There are no empirical results, which I think is ok given the analyses, though even simple experiments could further improve.
•	The paper adopts terminology and notation from Greenewald et al., but it should nonetheless be self-contained; for instance, q() shows up in Definition 2 without any description beyond being ‘a distribution over nodes’; `a weight of a branch’ not explained, what a ‘central node’ is and why it changes is confusing at first (the nodes themselves are not; eg topology is static). While the high-level search strategy is communicated (Fig 1 is helpful), I don’t think some of the important details of how that is done are.

Minor comments
•	line 6-7 ‘In this paper, we develop novel causal bandit algorithms without knowing the causal graph. Our algorithms work well for causal trees, causal forests and a general class of causal graphs.’  Slightly misleading, should qualify in abstract that the algorithm knows the essential graph.
•	For causal bandits, while it looks common to assume there is a single reward-generating variable, it sounds quite limiting.  The authors discuss this in part in Footnote 2 – the statement does not seem obvious to me though if they mean there is a single reward that depends on multiple variables.  If they mean that there are multiple observable rewards, each of which depends on a distinct single variable, then it is trivial.   If the former is meant, wouldn’t the reward function need to have quite special properties?
•	Minor issue, but line 140 discusses optimal intervention before the objective function introduced, line 143 uses Y which is not introduced, $X_t$ not defined
•	(very minor) Assumptions 2 and 3 – it is clear that both properties are needed, though I was a bit surprised that Assumption 2 does not imply Assumption 3 (though the dependence of \Delta on \epsilon, the graph topology, and P may be complicated).  I did not look at the appendix for the lower bound proof that contains an example, but if there is a simple high level explanation of why not, it might be worth mentioning.
•	(minor) Claim 1 is imprecise – what does it mean to be ‘determined’ or for the subtree to be ‘found’?  that there is a number of interventions that is sufficient for exact structure recovery  with high probability ?
•	(very minor) $B$ used for branch in graph and algorithm parameter (Theorem 1)
•	Lastly, I think some discussion, even high level, of per-round computational complexity would be beneficial.


**Time Spent Reviewing:**

10

---

> ### Author Response · Authors · 2021-08-10
> **Response to Reviewer xwY7**
>
> Thanks for your constructive comments. We are very glad you found our paper interesting. Please see below for our response.
>
> 1. Terminology and notation:
>
> a) In definition 2, $q()$ is used as the probability mass function over the tree nodes. So for q(B) where B is a branch of the tree, we mean $q(B) = \sum_{X\in V(B)} q(X)$, i.e. the total mass over all nodes in branch B. In our algorithm, $q_t()$ changes in different round $t$, so the central node regarding to $q_t()$ at each round also varies. We will explain this in detail and make sure our terminologies and notations are self-contained in our next version.
>
> b) In line 143, for $Y$ we mean the reward variable R. $X_t$ means the variable to be intervened at round $t$. Again thanks for pointing these out, we will correct them carefully.
>
>
> 2. Knowledge on essential graph:
>
> Like most of the intervention design works on causal discovery, we take the essential graph as the input of our algorithms. This is because the essential graph can be estimated accurately from enough observational data, which is much cheaper than interventional data and is usually not a concern in intervention design problems. See references below. We will mention this in the abstract, thanks for your suggestion.
>
> [1] He, Y.-B. and Geng, Z. (2008)
> Active learning of causal networks with intervention experiments and optimal designs.
> Journal of Machine Learning Research, 9(Nov):2523–2547.
>
> [2] Squires, C., Magliacane, S., Greenewald, K., Katz, D., Kocaoglu, M., and Shanmugam, K. (2020)
> Active structure learning of causal dags via directed clique tree.
> 34th Conference on Neural Information Processing Systems (NeurIPS 2020)
>
> [3] Greenewald, K., Katz, D., Shanmugam, K., Magliacane, S., Kocaoglu, M., Adsera, E. B., and Bresler, G. (2019)
> Sample efficient active learning of causal trees.
> 33rd Conference on Neural Information Processing Systems (NeurIPS 2019)
>
> 3. Multiple reward-generating variables:
>
> We have the following idea for the case where the reward has multiple direct causes (the ‘former’ one in your review).
> We will have to generalize Assumption 3 to hold on interventions on any ancestor of each of the direct causes. That is to say for any variable $X$ that is an ancestor of certain direct cause of the reward, we have $|E[R|do(X=x)] - E[R|do()]| > \Delta$ for some $x$ in the domain of $X$. Then our approach can be generalized to this setting by running multiple times.
> Specifically, in Algorithm 1, we don’t stop after stage 2 finds a reward generating variable, say $X_{R1}$, because it is only one of the direct causes.
> By construction of stage 2 (Algorithm 4), we know except for $X_{R1}$ itself, none of its descendants is a direct cause of the reward (we always check children first). We can fix the values of all variables in the subtree $\mathcal{T_1}$ induced by $X_{R1}$ (as the root).
> Then we re-run Algorithm 1 on the remaining graph (still a tree): original tree cut by subtree $\mathcal{T_1}$, and find a second direct cause of the reward. This procedure can be repeated until no further reward-generating variable can be found.
> Above idea is a way to find multiple reward generating variables, but it’s also possible to come up with more efficient ways. For example, instead of finding the direct causes one by one, it will be interesting to develop methods that can find several of them simultaneously. We think this setting itself is also interesting and worth studying as an independent work.
>
> 4. About Claim 1:
>
> For ‘determined’, we mean the learner will know whether $X$ is an ancestor of $X_R$ or $X$ is not an ancestor of $X_R$. For the directed subtree induced by $X$ as its root to be ‘found’, we mean the learner will be able to identify the true subtree whose root is $X$ in the original tree graph (includes all variables and edge directions in the subtree). We will make this claim more clear in our next version.
>
> 5. Other issues:
>
> a) About computational complexity:
> We provided sample complexity of interventions for stage 1 and 2 in Algorithm 1 (Lemma 1 in Appendix A.1). In stage 1 and 2, all computations for each intervention are for calculating interventional probabilities and updating the probabilities $q_t()$ that define the central node (see details in Algorithm 3 and 4). The computational complexity for Algorithm 1 in stage 1 and 2 is at most $\tilde{O}(K^2\max(1/\Delta^4, 1/\epsilon^4)d+n)$, ignoring poly-log terms, and stage 3 is just UCB. Causal forest and general graph settings are similar. We will add discussion on this.
>
> b) Whether Assumption 2 implies Assumption 3:
> No, Assumption 2 only describes the causal effect identifiability for adjacent variables on the causal graph D. Note that the reward variable is not contained in $D$. Assumption 3 describes the connection between the reward and its ancestors in $D$, so Assumption 2 does not imply Assumption 3.

---

### Comment · Area_Chair_RPF8 · 2021-09-01
**Questions to authors arising out of discussion**

Dear Authors,
      Thank you for your responses and engagement. There is a discussion amongst committee members that is on and there is some divergence in opinion.

 Some issues were raised. I thought it better to summarize some key issues here for you to quickly respond. Pls do respond to each of them even if you have otherwise responded similarly elsewhere before. I am partly quoting and partly paraphrasing issues that came up during the discussion.

a) Title suggests we start from an 'unknown graph'  while the first step requires having the full equivalence class graph available,

b) This unknown graph then needs to satisfy some extremely restrictive properties, where we can realistically only be sure these are satisfied if we actually know the full graph already, - in other words are intersection incompatibility assumption testable from the equivalence class or the graph ?

c) Reason for this extremely restrictive property is that for that specific subclass of models the proposed search strategy can be optimal, at the cost of failing on even very straightforward DAGs,

d) the claimed regret bounds in Thm2+3 are misleading in that an order N dependence is hidden in a constant that actually scales with N.
To be precise, for causal forests in general the constant C in Theorem 2 actually scales with N. Similarly for Theorem 3 where the number of chain components summed over also scales with N. That also implies that the subsequent claim that this bound shows it outperforms standard MAB algorithms is not supported. (Can u you clarify dependence of constant C and how it would compare against a standard MAB scaling ?)

e) Abstract seems a little misleading, in that it talks about general graphs, but the guarantees only apply to graphs with intersection incomparable property, which is a major limitation.

Can authors give quick responses to the above ?

Thanks

---

> ### Comment · Reviewer_sHAH · 2021-09-02
> **addition to Q to A**
>
> As a quick addition to the questions above:
>
> b) equivalence class could indeed be used to validate required structure for some graphs, but not for all graphs you currently can or want to handle. However the idea behind this suggestion by the chair would actually strengthen the paper quite a lot, in the sense that you CAN indeed validate your assumptions on the fly and see if they are met, and output 'no result' (or resort to another basic search strategy) if they are not. This is not what you currently do but should be easy to incorporate, and would remove the required strong limiting assumption on the structure.
>
> d) I know a worst case approach would scale with order N, so the statement here is not that your algorithm scales with N(log(N)^2), just that the current bound you derive implicitly does. As an example of this for causal forests (Theorem 2), consider a bipartite graph over N nodes, equally divided over the 2 layers. Then the reward X_R could be any node, but your algorithm has no means of exploiting the graph structure over the top layer (and so would on average have to try N/4 nodes, worst case N/2, if X_R was one of them). Similarly when X_R is in the bottom layer. In other words, apart from the tree case in Theorem 1, the regret bounds of your algorithm on causal forests in general still scale with N (like all methods), not with (log(N)^2).
>
> As a final separate remark I would encourage the authors to adapt the method to at least handle causal DAGs, while exploiting the central node approach developed in this paper for those parts of the search strategy where it applies. This would make the method actually useful in practice, while the regret bound of Theorem 1 would still apply if the model happened to satisfy the tree structure assumed here.

---

> > ### Author Response · Authors · 2021-09-02
> > **Response to Reviewer sHAH's addition to Q to A**
> >
> > We thank you for your suggestions!
> >
> > 1. We will incorporate the idea of validating required structures using the equivalent class. See our responses to b) and c) above for more details.
> >
> > 2. We note that in your example, the number of components is N/2, and $\Omega(N)$ interventions are unavoidable. We agree that for the *worst-cast causal forest*, our bound in Theorem 2 still scales with $N$. However, the bounds in Theorem 2 and Theorem 3 show our algorithm can *adapt* to benign cases (the number of chain components are much smaller than $N$) and achieve significantly better bounds than the worst case. See our responses to d) above for more details.
> >
> > 3. We note that it is straightforward to modify our Algorithm 2 to handle general DAG. If the intersection-incomparable assumption does not hold, Algorithm 2 may stop at line 5 or line 7 without finding any sub-junction-tree or clique that contains X_R for all components (note: Algo 2 *will not* output an incorrect sub-junction-tree or clique). If the learner finds that Algo 2 doesn’t output anything after line 5 or 7 for all components, they can use any standard MAB algorithm. See our responses to b) and c) above for more details. We will add discussions on how to handle general causal DAGs. Thanks again for your suggestions!

---

> > > ### Comment · Reviewer_sHAH · 2021-09-02
> > > **final remarks**
> > >
> > > 2) The claim in Theorem 2 is a regret bound, not a claim that there are conditions under which it works better than worst case. The given regret bound is misleading as for arbitrary causal forests the number of chain components scales with (constant) * N, with 'constant' somewhere between 0 and 1 depending on the sparsity and depth. But it still implies a term including N. So one workaround would be to drop Theorem 2(+3) claiming it is a bound, and merely use it as an example that it could work fast if the number of chain components is much smaller than N.
> > >
> > > Also the counterexample provided earlier applies to ALL bipartite graphs, irrespective of whether there are few or many nodes in the first layer or few or many connections between nodes in the two layers. In other words, for all bipartite graphs your algorithm has expected (so not even 'worst case') regret bound O(N), just like any standard MAB.
> > >
> > > 3) "If the learner finds that Algo 2 doesn’t output anything after line 5 or 7 for all components, they can use any standard MAB algorithm." => I am sorry but saying your algorithm can handle DAGs because if it doesn't work you can always use another algorithm that does is not good enough.

---

> > > > ### Author Response · Authors · 2021-09-02
> > > > **Response to your final remarks**
> > > >
> > > > 2. First, our bound is the standard form of regret bounds for structured bandits common in bandit literature. In particular, these bounds have a term that depends on the problem structure.
> > > > In our case, our regret depends on the graph structure, where $C$ denotes the number of components and $C$ can be smaller than $n$ in various graphs.
> > > > As long as the underlying graph has few components, our bounds can improve from standard MAB.
> > > >
> > > > We give two examples of regrets bounds for other structured bandit problems:
> > > >
> > > > *1)*  The regret bound for sparse linear bandits scales $O(\sqrt{dsT})$ where $d$ is the dimension and $s$ is the sparsity. Here, $s$ is also "number scales with (constant) *d, with 'constant' somewhere between 0 and 1".  See, e.g., Eq. (6) and its discussions in the paragraph below in
> > > > [1] Online-to-Confidence-Set Conversions and Application to Sparse Stochastic Bandits
> > > > Yasin Abbasi-Yadkori, David Pal, Csaba Szepesv´ari
> > > >
> > > > *2)* Other causal bandit works (that assume known causal graph) developed regret bound of $\tilde{O}(\sqrt{ZT})$ where $Z$ is a graph dependent number that is always $\le$ the action size and in the worst-case is the same of the action size. See, e.g.,
> > > > [2] Regret Analysis of Bandit Problems with Causal Background Knowledge
> > > > Yangyi Lu, Amirhossein Meisami, Ambuj Tewari, Zhenyu Yan
> > > >
> > > > We are happy to provide more clarifications about our bounds.
> > > >
> > > > Second,  we are confused why our bound scales with $O(N)$ for all bipartite graphs. Note you can view a chain graph of $2N$ nodes: $X_1\rightarrow X_2\rightarrow\cdots\rightarrow X_{2N}$ as a bipartite graph with $N$ nodes in each layer, say node 1,3,5,7... in layer one and node 2,4,6,8,...in layer two and node $i$ points to node $i+1$.
> > > > In this case, our regret bound scales with $O(\log N)$.
> > > > For this example, our algorithm *exploits the structure* (connections between nodes two layers) to obtain the improved bound.
> > > >
> > > >
> > > > 3. This is the most straightforward way to handle DAGs. However, we agree there can be a more elegant way of adapting our algorithm to handle DAGs. Thanks again for your suggestion!

---

> > > > > ### Comment · Reviewer_Em8x · 2021-09-03
> > > > > **response**
> > > > >
> > > > > Thank you everyone for the extended discussion. I think this was useful to further clarify some points and the pointers to the bandit literature by the authors for similar bounds are definitely useful.
> > > > >
> > > > > A minor suggestion I have based on the discussion above is to replace C in the bounds with C(D) or C(\varepsilon(D)) to make it clear that C is indeed not a constant but is a function of the graph or the essential component in consideration. Perhaps this may address parts of R#sHAH's concerns.

---

> > > > > > ### Author Response · Authors · 2021-09-03
> > > > > > **Response to reviewer Em8x**
> > > > > >
> > > > > > Thank you very much for your suggestion! We are happy to change our notations.

---

> ### Author Response · Authors · 2021-09-02
> **Response to questions to authors arising out of discussion**
>
> Thanks for raising these questions. Please see our responses below.
>
> **Response to a)** "Title suggests we start from an ’‘unknown graph’‘ while the first step requires having the full equivalence class graph available":
>
> We note that it’s been well-studied that the essential graph (full equivalence class graph) can be estimated accurately from enough observational data, e.g. using PC algorithm in [1].
> Compared to interventional data, observational data is much cheaper and is usually not a concern in intervention design problems (including our bandit setup).
> Many previous intervention design papers on causal discovery take the essential graph as the input of their algorithms because finding the equivalence class is not the bottleneck. See [2][3][4].
>
> [1] P. Spirtes, C. N. Glymour, and R. Scheines. Causation, Prediction, and Search. MIT Press, 2000.
>
> [2] He, Y.-B. and Geng, Z. (2008) Active learning of causal networks with intervention experiments and optimal designs. Journal of Machine Learning Research, 9(Nov):2523–2547.
>
> [3] Squires, C., Magliacane, S., Greenewald, K., Katz, D., Kocaoglu, M., and Shanmugam, K. (2020) Active structure learning of causal dags via directed clique tree. 34th Conference on Neural Information Processing Systems (NeurIPS 2020)
>
> [4] Greenewald, K., Katz, D., Shanmugam, K., Magliacane, S., Kocaoglu, M., Adsera, E. B., and Bresler, G. (2019) Sample efficient active learning of causal trees. 33rd Conference on Neural Information Processing Systems (NeurIPS 2019)
>
> **Response to b)** “This unknown graph then needs to satisfy some extremely restrictive properties, where we can realistically only be sure these are satisfied if we actually know the full graph already, - in other words are intersection incompatibility assumption testable from the equivalence class or the graph ?”
>
> **and** **c)** " Reason for this extremely restrictive property is that for that specific subclass of models the proposed search strategy can be optimal, at the cost of failing on even very straightforward DAGs"
>
> 1. As reviewer sHAH suggested, the equivalent class can be used to check whether the skeleton is indeed a tree or a forest. If not, we can use Algorithm 2.
>
> 2.  Algorithm 2 does not need to know whether the intersection-incomparable (denoted as I-I later)  assumption holds or not.
> I-I assumption guarantees that there’s no v-structure over cliques in any junction tree. So at each round, Algorithm 2 can eliminate at least half of the cliques by interventions over the central clique correctly.
> So under the I-I assumption, we can derive an improved regret bound in Theorem 3.
> Without this assumption, Algorithm 2 may stop at line 5 or line 7 without finding any sub-junction-tree or clique that contains X_R (note: Algo 2 *will not* output an incorrect sub-junction-tree or clique) for all components. If the learner finds that Algo 2 doesn’t output anything after line 5 or 7 for all components, they can use any standard MAB algorithm.
> Therefore, the algorithm guarantees the learner can always stop with $o(n)$ interventions before applying UCB algorithm in the last step.
>
> To summarize:
> If the learner does find $X_R$ or a clique that contains $X_R$, our algorithm has improved regret guarantees.
> If not, then the learner only uses $o(n)$ interventions to figure out that they need to turn to a standard MAB approach. Note that standard MAB methods need $T >= nK$ interventions to have an $\tilde{O}(nKT)$ regret. Running our algorithm for the first $o(n)$ interventions is not a high cost.
> We will add these discussions in the final version.
>
> 3.Lastly, we note that *I-I* assumption is not very restrictive because besides trees and forests, the graphs that satisfy *I-I* assumption also includes other classes of graphs such as proper interval graphs. See:
> Squires, C., Magliacane, S., Greenewald, K., Katz, D., Kocaoglu, M., and Shanmugam, K. (2020) Active structure learning of causal dags via directed clique tree. 34th Conference on Neural Information Processing Systems (NeurIPS 2020)
>
>
> **Response to d)** “the claimed regret bounds in Thm2+3 are misleading in that an order N dependence is hidden in a constant that actually scales with N”
>
> To be consistent with our paper, we use $n$ for the number of nodes below.
> For causal forest, the parameter $C$ denotes the number of chain components of the essential graph (i.e., the number of trees in the forest).
> *In the worst case*, all variables on the graph are isolated (this is also a causal forest where each tree has only one node), then $C = n$.  In this case, it is impossible for any causal algorithm to outperform standard MAB methods since no causal information can be exploited (we explained this issue in line 42-46; this was also proved in a lower bound result: Theorem 4 in [5]).
> This is similar to the example posed by reviewer sHAH.
>
> However, if the forest is composed of a few causal trees ($C<<n$), then Theorem 2 does improve the standard MAB methods. An extreme example is the causal tree setting where $C = 1$ and $n$ can be large.
>
> Same for Theorem 3, in the worst-case, the number of chain components can be $n$, but in the benign cases (some of the components contain many nodes), the number of components is much smaller than $n$.
>
> For comparison, standard MAB approaches always achieve $\tilde{O}(\sqrt{nKT})$.
> Our methods get rid of polynomial $n$ in benign cases and can easily direct the learner to use standard MAB approaches if some conditions do not hold (see our response to b) and c)).
> For example in causal forest, our method achieves $\tilde{O}(K(d+C)(log (n))^2+\sqrt{KT})$ regret (ignoring identifiability parameters and log-terms non-regarding to $n$). This bound improves significantly from the naive bounds when $n$ is large while $C$ is small.
> In Theorem 3, if there are fewer components and the clique numbers are small, the dominating term of the regret is $\tilde{O}(\sqrt{\omega(G_R)KT})$, where $\omega(G_R)$ denotes the clique number for the component G_R that contains $X_R$. $\omega(G_R)$ is always less than $n$. For the tree and forest setting, $\omega(G_R) = 2$.
>
> [5] Lu et al. (2020) Regret Analysis of Bandit Problems with Causal Background Knowledge (UAI 2020)
>
>
> **Response to e)** “Abstract seems a little misleading, in that it talks about general graphs, but the guarantees only apply to graphs with intersection incomparable property, which is a major limitation.”
>
> We are happy to change the sentence in the abstract,
> "Our algorithms work well for causal trees, causal forests and a general class of causal graphs"
>
> to something like:
>
> "Our algorithms give improved regret guarantees for causal trees, causal forests and a class of causal graphs that satisfy intersection-incomparable assumption, and degrade gracefully  to standard MAB approach when the causal graph does not have benign structures."
>
> We are also open to other suggestions.

---

### Decision · Program_Chairs · 2021-09-27

**Decision:**

Accept (Poster)

**Comment:**

All reviews are extremely favorable except one dissenting review.

This paper produced a very long discussion and engagement between dissenting reviewer, other reviewers and authors. I would like to thank all reviewers (especially the dissenting reviewer) for participating in the discussion.

I would like to state the main points of the dissenting review  and I feel that the discussion and authors final responses does address most of the serious concerns raised. Therefore, I am recommending acceptance.

Particularly, this is the first paper (I am aware of) that solves regret minimization problem for causal bandits when only the equivalence class is known. The authors leverage existing structure learning results to obtain an algorithm that ensures non trivial regret (non trivial regret guarantee holds under some technical conditions).

Main points from the dissenting review - followed by paraphrased answers from either the authors/other reviewers.

a) First step requires having the full equivalence class graph available

A) many other works in active structure learning literature makes these assumptions. This paper considers a harder problem of regret minimization without full structure knowledge, this would be not be a major penalizing factor. Besides, interventionally it does not cost anything to assume one has enough observational samples. Therefore, it is a resonable assumption.

b) "unknown graph then needs to satisfy some extremely restrictive properties"

A)  Authors assume intersection incomparable property on the clique tree decomposition of the observational markov equivalence class. The property assumed can be tested with only the equivalence class which is an input to the bandit.

 Nevertheless, this is a popular assumption - both in structure learning and some prior literature. Another reviewer even hinted that in a random graph, this may be satisfied even with high probability (or at least depends on random graph model assumptions). This is satisfied with popular chordal graph sub families.

Other aspect is that the bandit algorithm would run irrespective of the assumption. It would halt at a specific line and then the bandit part of the algorithm would commence with potentially large number of arms.

*Note to authors*: I would urge the authors to note this point explicitly in camera ready and provide a complete algorithm with the halting conditions etc...

Not withstanding the above, it is an assumption to avoid a 'pathology' (in my opinion) in the clique tree.

c) The claimed regret bounds in Thm2+3 are misleading in that an order N dependence is hidden in a constant that actually scales with N

A) All regret bounds in MAB problems scale with problem parameters - ambient dimension in linear bandits, number of arms in multi armed bandits, some graph parameter when DAG is known in causal bandits. Here, this is number of chain components in the chordal graph which is a parameter that depends on the EC. If all nodes are isolated this would reduce to standard MAB and therefore such a regret scaling would be unavoidable.